# Robust Offline Active Learning on Graphs

**Yuanchen Wu**
Department of Statistics
The Pennsylvania State University
yqw5734@psu.edu

**Yubai Yuan**
Department of Statistics
The Pennsylvania State University
yvy5509@psu.edu

## Abstract

We consider the problem of active learning on graphs for node-level tasks, which has crucial applications in many real-world networks where labeling node responses is expensive. In this paper, we propose an offline active learning method that selects nodes to query by explicitly incorporating information from both the network structure and node covariates. Building on graph signal recovery theories and the random spectral sparsification technique, the proposed method adopts a two-stage biased sampling strategy that takes both *informativeness* and *representativeness* into consideration for node querying. *Informativeness* refers to the complexity of graph signals that are learnable from the responses of queried nodes, while *representativeness* refers to the capacity of queried nodes to control generalization errors given noisy node-level information. We establish a theoretical relationship between generalization error and the number of nodes selected by the proposed method. Our theoretical results demonstrate the trade-off between *informativeness* and *representativeness* in active learning. Extensive numerical experiments show that the proposed method is competitive with existing graph-based active learning methods, especially when node covariates and responses contain noises. Additionally, the proposed method is applicable to both regression and classification tasks on graphs.

## 1 Introduction

In many graph-based semi-supervised learning tasks for node-level prediction, labeled nodes are scarce, and the labeling process often incurs high costs in real-world applications. Randomly sampling nodes for labeling can be inefficient, as it overlooks label dependencies across the network. Active learning [29] addresses this issue by selecting informative nodes for labeling by human annotators, thereby improving the performance of downstream prediction algorithms.

Active learning is closely related to the optimal experimental design principle [36] in statistics. Traditional optimal experimental design methods select samples to maximize a specific statistical criterion [26, 19]. However, these methods often are not designed to incorporate network structure, therefore inefficient for graph-based learning tasks. On the other hand, selecting informative nodes on a network is studied extensively in the graph signal sampling literature [11, 18, 10, 28]. These strategies are typically based on the principle of *network homophily*, which assumes that connected nodes tend to have similar labels. However, a node's label often also depends on its individual covariates. Therefore, signal-sampling strategies that focus solely on network information may miss critical insights provided by covariates.

Recently, inspired by the great success of graph neural networks (GNNs) [16, 34] in graph-based machine learning tasks, many GNN-based active learning strategies have been proposed. Existing methods select nodes to query by maximizing *information gain* under different criteria, including information entropy [4], the number of influenced nodes [39], prediction uncertainty [22], expected error reduction [8], and expected model change [30]. Most of these information gain measurements

are defined in the spatial domain, leveraging the message-passing framework of GNNs to incorporate both network structure and covariate information. However, their effectiveness in maximizing learning outcomes is not guaranteed and can be difficult to evaluate. This challenge arises from the difficulty of quantifying node labeling complexity in the spatial domain due to intractable network topologies. While complexity measures exist for binary classification over networks [9], their extension to more complex graph signals incorporating node covariates remains unclear. This lack of well-defined complexity measures complicates performance analysis and creates a misalignment between graph-based information measurements and the gradient used to search the labeling function space, potentially leading to sub-optimal node selection.

Moreover, from a practical perspective, most of the previously discussed methods operate in an online setting, requiring prompt labeling feedback from an external annotator. However, this online framework is not always feasible when computational resources are limited [24] or when recurrent interaction between the algorithm and the annotator is impractical, such as in remote sensing or online marketing tasks [32, 35]. Additionally, both network data and annotator-provided labels may contain measurement errors. These methods often fail to account for noise in the training data [23], which can significantly degrade the prediction performance of models on unlabeled nodes [7, 21].

To address these challenges, we propose an offline active learning on graphs framework for node-level prediction tasks. Inspired by the theory of graph signal recovery [11, 18, 28] and GNNs, we first introduce a graph function space that integrates both node covariate information and network topology. The complexity of the node labeling function within this space is well-defined in the graph spectral domain. Accordingly, we propose a query information gain measurement aligned with the spectral-based complexity, allowing our strategy to achieve theoretically optimal sample complexity.

Building on this, we develop a greedy node query strategy. The labels of the queried nodes help identify orthogonal components of the target labeling function, each with varying levels of smoothness across the network. To address data noise, the query procedure considers both *informativeness*—the contribution of queried nodes in recovering non-smooth components of a signal—and *representativeness*—the robustness of predictions against noise in the training data. Compared to existing methods, the proposed approach provides a provably effective strategy under general network structures and achieves higher query efficiency by incorporating both network and node covariate information.

The proposed method identifies the labeling function via a bottom-up strategy—first identifying the smoother components of the labeling function and then continuing to more oscillated components. Therefore, the proposed method is naturally robust to high-frequency noise in node covariates. We provide a theoretical guarantee for the effectiveness of the proposed method in semi-supervised learning tasks. The generalization error bound is guaranteed even when the node labels are noisy. Our theoretical results also highlight an interesting trade-off between informativeness and representativeness in graph-based active learning.

## 2 Preliminaries

We consider an undirected, weighted, connected graph $\mathbf{G} = \{\mathbf{V}, \mathbf{A}\}$, where $\mathbf{V} = \{1, 2, \cdots, n\}$ is the set of $n$ nodes, and $\mathbf{A} \in \mathbb{R}^{n \times n}$ is the symmetric adjacency matrix, with element $a_{ij} \geq 0$ denoting the edge weight between nodes $i$ and $j$. The degree matrix is defined as $\mathbf{D} = \text{diag}\{d_1, d_2, \cdots, d_n\}$, where $d_i = \sum_{1 \leq i \leq n} a_{ij}$ denotes the degree of node $i$. Additionally, we observe the node response vector $\mathbf{Y} \in \mathbb{R}^{n \times 1}$ and the node covariate matrix $\mathbf{X} = (X_1, \cdots, X_p) \in \mathbb{R}^{n \times p}$, where the $i^{th}$ row, $\mathbf{X}_{i\cdot}$, is the $p$-dimensional covariate vector for node $i$. The linear space of all linear combinations of $\{X_1, \cdots, X_p\}$ is denoted as $\text{Span}\{X_1, \cdots, X_p\}$. The normalized graph Laplacian matrix is defined as $\mathcal{L} = \mathbf{I} - \mathbf{D}^{-1/2} \mathbf{A} \mathbf{D}^{-1/2}$, where $\mathbf{I}$ is the $n \times n$ identity matrix. The matrix $\mathcal{L}$ is symmetric and positive semi-definite, with $n$ real eigenvalues satisfying $0 = \lambda_1 \leq \lambda_2 \leq \cdots \leq \lambda_n \leq 2$, and a corresponding set of eigenvectors denoted by $\mathbf{U} = \{U_1, U_2, \cdots, U_n\}$. We use $b = \mathcal{O}(a)$ to indicate $|b| \leq M|a|$ for some $M > 0$. For a set of nodes $\mathcal{S} \subset \mathbf{V}$, $|\mathcal{S}|$ indicates its cardinality, and $S^c = \mathbf{V} \backslash \mathcal{S}$ denotes the complement of $\mathcal{S}$.

### 2.1 Graph signal representation

Consider a graph signal $\mathbf{f} \in \mathbb{R}^n$, where $\mathbf{f}(i)$ denotes the signal value at node $i$. For a set of nodes $S$, we define the subspace $\mathbf{L}_{\mathcal{S}} := \{\mathbf{f} \in \mathbb{R}^n \mid \mathbf{f}(S^c) = 0\}$, where $\mathbf{f}(S) \in \mathbb{R}^{|\mathcal{S}|}$ represents the values of $\mathbf{f}$

on nodes in $\mathcal{S}$. In this paper, we consider both regression tasks, where $\mathbf{f}(i)$ is a continuous response, and classification tasks, where $\mathbf{f}(i)$ is a multi-class label.

Since $\mathbf{U}$ serves as a set of bases for $\mathbb{R}^n$, we can decompose $\mathbf{f}$ in the graph spectral domain as $\mathbf{f} = \sum_{j=1}^{n} \alpha_{\mathbf{f}}(\lambda_j) U_j$, where $\alpha_{\mathbf{f}}(\lambda_j) = \langle \mathbf{f}, U_j \rangle$ is defined as the graph Fourier transform (GFT) coefficient corresponding to frequency $\lambda_j$. From a graph signal processing perspective, a smaller eigenvalue $\lambda_k$ indicates lower variation in the associated eigenvector $U_k$, reflecting smoother transitions between neighboring nodes. Therefore, the smoothness of $\mathbf{f}$ over the network can be characterized by the magnitude of $\alpha_{\mathbf{f}}(\lambda_j)$ at each frequency $\lambda_j$. More formally, we measure the signal complexity of $\mathbf{f}$ using the bandwidth frequency $\omega_{\mathbf{f}} = \sup\{\lambda_j | \alpha_{\mathbf{f}}(\lambda_j) > 0\}$. Accordingly, we define the subspace of graph signals with a bandwidth frequency less than or equal to $\omega$ as $\mathbf{L}_\omega := \{\mathbf{f} \in \mathbb{R}^n \mid \omega_{\mathbf{f}} \leq \omega\}$. It follows directly that $\forall \omega_1 < \omega_2, \mathbf{L}_{\omega_1} \subset \mathbf{L}_{\omega_2}$.

## 2.2 Active semi-supervised learning on graphs

The key idea in graph-based semi-supervised learning is to reconstruct the graph signal $\mathbf{f}$ within a function space $\mathbf{H}_\omega(\mathbf{X}, \mathbf{A})$ that depends on both the network structure and node-wise covariates, where the frequency parameter $\omega$ controls the size of the space to mitigate overfitting. Assume that $Y_i$ is the observed noisy realization of the true signal $\mathbf{f}(i)$ at node $i$, active learning operates in a scenario where we have limited access to $Y_i$ on only a subset of nodes $\mathcal{S}$, with $|\mathcal{S}| << n$. The objective is to estimate $\mathbf{f}$ within $\mathbf{H}_\omega(\mathbf{X}, \mathbf{A})$ using $\{Y_i\}_{i \in \mathcal{S}}$ by considering the empirical estimator of $\mathbf{f}$ as

$$\mathbf{f}_\mathcal{S} = \underset{\mathbf{g} \in \mathbf{H}_\omega(\mathbf{X}, \mathbf{A})}{\arg \min} \sum_{i \in \mathcal{S}} l\big(Y_i, \mathbf{g}(i)\big), \tag{1}$$

where $l(\cdot)$ is a task-specific loss function. We denote $\mathbf{f}^*$ as the minimizer of (1) when responses on all nodes are available, i.e., $\mathbf{f}^* = \mathbf{f_V}$. The goal of active semi-supervised learning is to design an appropriate function space $\mathbf{H}_\omega(\mathbf{X}, \mathbf{A})$ and select an *informative* subset of nodes $\mathcal{S}$ for querying responses, under the query budget $|\mathcal{S}| \leq \mathcal{B}$, such that the estimation error is bounded as follows:

$$\|\mathbf{f}_\mathcal{S} - \mathbf{f}^*\|_2^2 \leq \rho \|\mathbf{f}^* - \mathbf{f}\|_2^2$$

For a fixed $\mathcal{B}$, we wish to minimize the parameter $\rho > 0$, which converges to $0$ as the query budget $\mathcal{B}$ approaches $n$.

# 3 Biased Sequential Sampling

In this section, we introduce a function space for recovering the graph signal. Leveraging this function space, we propose an offline node query strategy that integrates criteria of both node *informativeness* and *representativeness* to infer the labels of unannotated nodes in the network.

## 3.1 Graph signal function space

In semi-supervised learning tasks on networks, both the network topology and node-wise covariates are crucial for inferring the graph signal. To effectively incorporate this information, we propose a function class for reconstructing the graph signal that lies at the intersection of the graph spectral domain and the space of node covariates. Motivated by the graph Fourier transform, we define the following function class:

$$\mathbf{H}_\omega(\mathbf{X}, \mathbf{A}) = \text{Proj}_{\mathbf{L}_\omega} \text{Span}(\mathbf{X}) := \text{Span}\{\text{Proj}_{\mathbf{L}_\omega} X_1, \cdots, \text{Proj}_{\mathbf{L}_\omega} X_p\},$$

$$\text{where } \text{Proj}_{\mathbf{L}_\omega} X_i = \sum_{j:\lambda_j \leq \omega} \langle X_i, U_j \rangle U_j.$$

Here, the choice of $\omega$ balances the information from node covariates and network structure. When $\omega = 2$, $\mathbf{H}_\omega(\mathbf{X}, \mathbf{A})$ spans the full column space of covariates, i.e., $\text{Span}\{X_1, \cdots, X_p\}$, allowing for a full utilization of the original covariate space to estimate the graph signal, but without incorporating any network information. On the other hand, when $\omega$ is close to zero—consider, for example, the extreme case where $|\{U_j \mid \lambda_j \leq \omega\}| = 2$ and $p \gg 2$—then $\text{Proj}_{\mathbf{L}_\omega} X_i$ reduces to $\text{Span}\{U_1, U_2\}$, resulting in a loss of critical information provided by the original $\mathbf{X}$.

By carefully choosing $\omega$, however, this function space can offer two key advantages for estimating the graph signal. From a signal recovery perspective, $\mathbf{H}_\omega(\mathbf{X}, \mathbf{A})$ imposes graph-based regularization

over node covariates, enhancing generalizability when the dimension of covariates $p$ exceeds the query budget or even the network size—conditions commonly encountered in real applications. Additionally, covariate smoothing filters out signals in the covariates that are irrelevant to network-based prediction, thereby increasing robustness against potential noise in the covariates. From an active learning perspective, using $\mathbf{H}_\omega(\mathbf{X}, \mathbf{A})$ enables a bottom-up query strategy that begins with a small $\omega$ to capture the smoothest global trends in the graph signal. As the labeling budget increases, $\omega$ is adaptively increased to capture more complex graph signals within the larger space $\mathbf{H}_\omega(\mathbf{X}, \mathbf{A})$.

The graph signal $\mathbf{f}$ can be approximated by its projection onto the space $\mathbf{H}_\omega(\mathbf{X}, \mathbf{A})$. Specifically, let $\mathbf{U}_d = \{U_1, U_2, \cdots, U_d\} \in \mathbb{R}^{n \times d}$ stack the $d$ leading eigenvectors of $\mathcal{L}$, where $d = \arg\max_{1 \le j \le n}(\lambda_j - \omega) \le 0$. The graph signal estimation is then given by $\mathbf{U}_d \mathbf{U}_d^T \mathbf{X}\beta$, where $\beta \in \mathbb{R}^d$ is a trainable weight vector. However, the parameters $\beta$ may become unidentifiable when the covariate dimension $p$ exceeds $d$. To address this issue, we reparameterize the linear regression as follows:

$$\mathbf{U}_d \mathbf{U}_d^T \mathbf{X}\beta = \tilde{\mathbf{X}}\tilde{\beta}, \tag{2}$$

where $\tilde{\beta} = \Sigma V_2^T \beta$ and $\tilde{\mathbf{X}} = \mathbf{U}_d V_1$. Here, $V_1 \in \mathbb{R}^{d \times r}$, $V_2 \in \mathbb{R}^{p \times r}$, and $\Sigma \in \mathbb{R}^{r \times r}$ denote the left and right singular vectors and the diagonal matrix of the $r$ singular values, respectively.

In the reparameterized form (2), the columns of $\tilde{\mathbf{X}}$ serve as bases for $\mathbf{H}_\omega(\mathbf{X}, \mathbf{A})$, thus $\dim(\mathbf{H}_\omega(\mathbf{X}, \mathbf{A})) = \text{rank}(\tilde{\mathbf{X}}) = r \le \min\{d, p\}$. The transformed predictors $\tilde{\mathbf{X}}$ capture the components of the node covariates constrained within the low-frequency graph spectrum. A graph signal $\mathbf{f} \in \mathbf{H}_\omega(\mathbf{X}, \mathbf{A})$ can be parameterized as a linear combination of the columns of $\tilde{\mathbf{X}}$, with the corresponding weights $\tilde{\beta}$ identified via

$$\hat{\beta} = \arg\min_{\tilde{\beta}} \sum_{i \in \mathcal{S}} \left(\mathbf{f}(i) - (\tilde{\mathbf{X}}_\mathbf{S})_{i\cdot}\, \tilde{\beta}\right)^2 \tag{3}$$

where $\tilde{\mathbf{X}}_\mathbf{S} \in \mathrm{R}^{|\mathcal{S}| \times r}$ is the submatrix of $\tilde{\mathbf{X}}$ containing rows indexed by the query set $\mathcal{S}$, and $\{\mathbf{f}(i)\}_{i \in \mathcal{S}}$ represents the true labels for nodes in $\mathcal{S}$. To achieve the identification of $\mathbf{f}$, it is necessary that $|\mathcal{S}| \ge r$; otherwise, there will be more parameters than equations in (3). More importantly, since $\text{rank}(\tilde{\mathbf{X}}_\mathbf{S}) \le \text{rank}(\tilde{\mathbf{X}}) = r$, $\mathbf{f}$ is only identifiable if $\tilde{\mathbf{X}}_\mathbf{S}$ has full column rank. Notice that $r$ increases monotonically with $\omega$. If $\mathcal{S}$ is not carefully selected, the graph signal can only be identified in $\mathbf{H}_{\omega'}(\mathbf{X}, \mathbf{A})$ for some $\omega' < \omega$, which is a subspace of $\mathbf{H}_\omega(\mathbf{X}, \mathbf{A})$.

### 3.2 Informative node selection

We first define the identification of $\mathbf{H}_\omega(\mathbf{X}, \mathbf{A})$ by the node query set $\mathcal{S}$ as follows:

**Definition 1** *A subset of nodes $\mathcal{S} \subset \mathbf{V}$ can identify the graph signal space $\mathbf{H}_\omega(\mathbf{X}, \mathbf{A})$ up to frequency $\omega$ if, for any two functions $\mathbf{f}_1, \mathbf{f}_2 \in \mathbf{H}_\omega(\mathbf{X}, \mathbf{A})$ such that $\mathbf{f}_1(i) = \mathbf{f}_2(i)$ for all $i \in \mathcal{S}$, it follows that $\mathbf{f}_1(j) = \mathbf{f}_2(j)$ for all $j \in \mathbf{V}$.*

Intuitively, the informativeness of a set $\mathcal{S}$ can be quantified by the frequency $\omega$ corresponding to the space $\mathbf{H}_\omega(\mathbf{X}, \mathbf{A})$ that can be identified. To select informative nodes, we need to bridge the query set $\mathcal{S}$ in the spatial domain with $\omega$ in the spectral domain. To achieve this, we consider the counterpart of the function space $\mathbf{H}_\omega(\mathbf{X}, \mathbf{A})$ in the spatial domain. Specifically, we introduce the projection space with respect to a subset of nodes $\mathcal{S}$ as follows: $\mathbf{H}_\mathcal{S}(\mathbf{X}, \mathbf{A}) := \text{Span}\{X_1^{(\mathcal{S})}, \cdots, X_p^{(\mathcal{S})}\}$, where

$$X_p^{(\mathcal{S})}(i) = \begin{cases} X_p(i) & \text{if } i \in \mathcal{S} \\ \mathbf{0} & \text{if } i \in \mathcal{S}^c \end{cases}$$

Here, $X_p(i)$ denotes the $p$-th covariate for node $i$ in $\mathbf{X}$. Theorem 3.1 establishes a connection between the two graph signal spaces $\mathbf{H}_\omega(\mathbf{X}, \mathbf{A})$ and $\mathbf{H}_\mathcal{S}(\mathbf{X}, \mathbf{A})$, providing a metric for evaluating the informativeness of querying a subset of nodes on the graph.

**Theorem 3.1** *Any graph signal $\mathbf{f} \in \mathbf{H}_\omega(\mathbf{X}, \mathbf{A})$ can be identified using labels on a subset of nodes $\mathcal{S}$ if and only if:*

$$\omega < \omega(\mathcal{S}) := \inf_{\mathbf{g} \in \mathbf{H}_{\mathcal{S}^c}(\mathbf{X}, \mathbf{A})} \omega_\mathbf{g}, \tag{4}$$

*where $\mathcal{S}^c$ denotes the complement of $\mathcal{S}$ in $\mathbf{V}$.*

We denote the quantity $\omega(\mathcal{S})$ in (4) as the *bandwidth frequency* with respect to the node set $\mathcal{S}$. This quantity can be explicitly calculated and measures the size of the space $\mathbf{H}_\omega(\mathbf{X}, \mathbf{A})$ that can be recovered from the subset of nodes $\mathcal{S}$. The goal of the active learning strategy is to select $\mathcal{S}$ within a given budget to maximize the bandwidth frequency $\omega(\mathcal{S})$, thus enabling the identification of graph signals with the highest possible complexity.

To calculate the bandwidth frequency $\omega(\mathcal{S})$, consider any graph signal $\mathbf{g}$ and its components with non-zero frequency $\Lambda_\mathbf{g} := \{\lambda_i \mid \alpha_\mathbf{g}(\lambda_i) > 0\}$. We use the fact that

$$\lim_{k \to \infty} \left( \sum_{j:\lambda_j \in \Lambda_\mathbf{g}} c_j \lambda_j^k \right)^{1/k} = \max_{\lambda_j \in \Lambda_\mathbf{g}} (\lambda_j),$$

where $\sum_{j:\lambda_j \in \Lambda_\mathbf{g}} c_j = 1$ and $0 \leq c_j \leq 1$. Combined with the Rayleigh quotient representation of eigenvalues, the bandwidth frequency $\omega_\mathbf{g}$ can be calculated as

$$\omega_\mathbf{g} = \lim_{k \to \infty} \omega_\mathbf{g}(k), \quad \text{where} \quad \omega_\mathbf{g}(k) = \left( \frac{\mathbf{g}^T \mathcal{L}^k \mathbf{g}}{\mathbf{g}^T \mathbf{g}} \right)^{1/k}.$$

As a result, we can approximate the bandwidth $\omega_\mathbf{g}$ using $\omega_\mathbf{g}(k)$ for a large $k$. Maximizing $\omega(\mathcal{S})$ over $\mathcal{S}$ then transforms into the following optimization problem:

$$\mathcal{S} = \arg\max_{\mathcal{S}:|\mathcal{S}| \leq \mathcal{B}} \hat{\omega}(\mathcal{S}), \quad \text{where} \quad \hat{\omega}(\mathcal{S}) := \inf_{\mathbf{g} \in \mathbf{H}_{\mathcal{S}^c}(\mathbf{X}, \mathbf{A})} \omega_\mathbf{g}^k(k), \tag{5}$$

where $\mathcal{B}$ represents the budget for querying labels. Due to the combinatorial complexity of directly solving optimization problem (5) by simultaneously selecting $\mathcal{S}$, we propose a greedy selection strategy as a continuous relaxation of (5).

The selection procedure starts with $\mathcal{S} = \emptyset$ and sequentially adds one node to $\mathcal{S}$ that maximizes the increase in $\omega(\mathcal{S})$ until the budget is reached. We introduce an $n$-dimensional vector $\mathbf{t} = (t_1, t_2, \cdots, t_n)^T$ with $0 \leq t_i \leq 1$, and define the corresponding diagonal matrix $D(\mathbf{t})$ with diagonal entries given by $\mathbf{t}$. This allows us to encode the set of query nodes using $\mathbf{t} = \mathbf{1}_\mathcal{S}$, where $\mathbf{1}_\mathcal{S}(i) = 1$ if $i \in \mathcal{S}$ and $\mathbf{1}_\mathcal{S}(i) = 0$ if $i \in \mathcal{S}^c$. We then consider the space spanned by the columns of $D(\mathbf{t})\mathbf{X}$ as $\text{Span}\{D(\mathbf{t})\mathbf{X}\}$, and the following relation holds:

$$\mathbf{H}_{\mathcal{S}^c}(\mathbf{X}, \mathbf{A}) = \text{Span}\{D(\mathbf{1}_{\mathcal{S}^c})\mathbf{X}\}.$$

Intuitively, $\text{Span}\{D(\mathbf{t})\mathbf{X}\}$ acts as a differentiable relaxation of the subspace $\mathbf{H}_\mathcal{S}(\mathbf{X}, \mathbf{A})$, enabling perturbation analysis of the bandwidth frequency when a new node is added to $\mathcal{S}$. The projection operator associated with $\text{Span}\{D(\mathbf{t})\mathbf{X}\}$ can be explicitly expressed as

$$\mathbf{P}(\mathbf{t}) = D(\mathbf{t})\mathbf{X} \left( \mathbf{X}^T D(\mathbf{t}^2)\mathbf{X} \right)^{-1} \mathbf{X}^T D(\mathbf{t}).$$

To quantify the increase in $\hat{\omega}(\mathcal{S})$ when adding a new node to $\mathcal{S}$, we consider the following regularized optimization problem:

$$\lambda_\alpha(\mathbf{t}) = \min_\phi \frac{\phi^T \mathcal{L}^k \phi}{\phi^T \phi} + \alpha \frac{\phi^T (\mathbf{I} - \mathbf{P}(\mathbf{t})) \phi}{\phi^T \phi}. \tag{6}$$

The penalty term on the right-hand side of (6) encourages the graph signal $\phi$ to remain in $\mathbf{H}_{\mathcal{S}^c}(\mathbf{X}, \mathbf{A})$. As the parameter $\alpha$ approaches infinity and $\mathbf{t} = \mathbf{1}_{\mathcal{S}^c}$, the minimization $\lambda_\alpha(\mathbf{1}_{\mathcal{S}^c})$ in (6) converges to $\hat{\omega}(\mathcal{S})$ in (5). The information gain from labeling a node $i \in \mathcal{S}^c$ can then be quantified by the gradient of the bandwidth frequency as $t_i$ decreases from 1 to 0:

$$\Delta_i := -\frac{\partial \lambda_\alpha(\mathbf{t})}{\partial t_i}\bigg|_{\mathbf{t} = \mathbf{1}_{\mathcal{S}^c}} = 2\alpha \times \phi^T \frac{\partial \mathbf{P}(\mathbf{t})}{\partial t_i} \phi \bigg|_{\mathbf{t} = \mathbf{1}_{\mathcal{S}^c}}, \tag{7}$$

where $\phi$ is the minimizer of (6) at $\mathbf{t} = \mathbf{1}_{\mathcal{S}^c}$, which corresponds to the eigenvector associated with the smallest non-zero eigenvalue of the matrix $\mathbf{P}(\mathbf{1}_{\mathcal{S}^c}) \mathcal{L}^k \mathbf{P}(\mathbf{1}_{\mathcal{S}^c})$. We then select the node $i = \arg\max_{j \in \mathcal{S}^c} \Delta_j$ and update the query set as $\mathcal{S} = \mathcal{S} \cup \{i\}$. The explicit representation of $\Delta_i$ in (7) can be found in Appendix B.5.

## 3.3 Representative node selection

In real-world applications, we often have access only to a perturbed version of the true graph signals, denoted as $Y = \mathbf{f} + \xi$, where $\xi$ represents node labeling noise that is independent of the network data. When replacing the true label $\mathbf{f}(i)$ with $Y(i)$ in (3), this noise term introduces both finite-sample bias and variance in the estimation of the graph signal $\mathbf{f}$. As a result, we aim to query nodes that are sufficiently *representative* of the entire covariate space to bound the generalization error. To achieve this, we introduce principled randomness into the deterministic selection procedure described in Section 3.2 to ensure that $\mathcal{S}$ includes nodes that are both informative and representative. The modified graph signal estimation procedure is given by:

$$\hat{\beta} = \arg\min_{\tilde{\beta}} \sum_{i \in \mathcal{S}} s_i \big(Y(i) - (\tilde{\mathbf{X}}_{\mathbf{S}})_{i\cdot}\, \tilde{\beta}\big)^2, \tag{8}$$

where $s_i$ is the weight associated with the probability of selecting node $i$ into $\mathcal{S}$.

Specifically, the generalization error of the estimator in (8) is determined by the smallest eigenvalue of $\tilde{\mathbf{X}}_{\mathcal{S}}^T \tilde{\mathbf{X}}_{\mathcal{S}}$, denoted as $\lambda_{\min}(\tilde{\mathbf{X}}_{\mathcal{S}}^T \tilde{\mathbf{X}}_{\mathcal{S}})$. Given that $\lambda_{\min}(\tilde{\mathbf{X}}^T \tilde{\mathbf{X}}) = 1$, our goal is to increase the representativeness of $\mathcal{S}$ such that $\lambda_{\min}(\tilde{\mathbf{X}}_{\mathcal{S}}^T \tilde{\mathbf{X}}_{\mathcal{S}})$ is lower-bounded by:

$$\lambda_{\min}\left(\tilde{\mathbf{X}}_{\mathcal{S}}^T \tilde{\mathbf{X}}_{\mathcal{S}}\right) \geq (1 - o_{|\mathcal{S}|}(1))\lambda_{\min}\left(\tilde{\mathbf{X}}^T \tilde{\mathbf{X}}\right). \tag{9}$$

However, the informative selection method in Section 3.2 does not guarantee (9). To address this, we propose a sequential biased sampling approach that balances informative node selection with generalization error control.

The key idea to achieve a lower bound for $\lambda_{\min}(\tilde{\mathbf{X}}_{\mathcal{S}}^T \tilde{\mathbf{X}}_{\mathcal{S}})$ is to use spectral sparsification techniques for positive semi-definite matrices [15]. Let $v_i \in \mathbb{R}^{1 \times r}$ denote the $i$-th row of the constrained basis $\tilde{\mathbf{X}}$. By definition of $\tilde{\mathbf{X}}$, it follows that $\mathbf{I}_{r \times r} = \sum_{i=1}^n v_i^T v_i$. Inspired by the randomized sampling approach in [17], we propose a biased sampling strategy to construct $\mathcal{S}$ with $|\mathcal{S}| \ll n$ and weights $\{s_i > 0, i \in \mathcal{S}\}$ such that $\sum_{i \in \mathcal{S}} s_i v_i^T v_i \approx \mathbf{I}$. In other words, the weighted covariance matrix of the query set $\mathcal{S}$ satisfies $\lambda_{\min}(\tilde{\mathbf{X}}_{\mathcal{S}}^T W_S \tilde{\mathbf{X}}_{\mathcal{S}}) \approx 1$, where $W_S$ is a diagonal matrix with $s_i$ on its diagonal.

We outline the detailed sampling procedure as follows. After the $(t-1)^{\text{th}}$ selection, let the set of query nodes be $\mathcal{S}_{t-1}$ with corresponding node-wise weights $\mathcal{W}_{t-1} = \{s_j > 0 \mid j \in \mathcal{S}_{t-1}\}$. The covariance matrix of $\mathcal{S}_{t-1}$ is given by $C_{t-1} \in \mathbb{R}^{r \times r}$, defined as $C_{t-1} = \tilde{\mathbf{X}}_{\mathcal{S}_{t-1}}^T \tilde{\mathbf{X}}_{\mathcal{S}_{t-1}} = \sum_{j \in \mathcal{S}_{t-1}} s_j v_j^T v_j$. To analyze the behavior of eigenvalues as the query set is updated, we follow [17] and introduce the potential function:

$$\Phi_{t-1} = \text{Tr}[(u_{t-1}I - C_{t-1})^{-1}] + \text{Tr}[(C_{t-1} - l_{t-1}I)^{-1}], \tag{10}$$

where $u_{t-1}$ and $l_{t-1}$ are constants such that $l_{t-1} < \lambda_{\min}(C_{t-1}) \leq \lambda_{\max}(C_{t-1}) < u_{t-1}$, and $\text{Tr}(\cdot)$ denotes the trace of a matrix. The potential function $\Phi_{t-1}$ quantifies the coherence among all eigenvalues of $C_{t-1}$. To construct the candidate set $B_m$, we sample node $i$ and update $C_t$, $u_t$, and $l_t$ such that all eigenvalues of $C_t$ remain within the interval $(l_t, u_t)$. To achieve this, we first calculate the node-wise probabilities $\{p_i\}_{i=1}^n$ as:

$$p_i = \left[v_i(u_{t-1}I - C_{t-1})^{-1}v_i^T + v_i(C_{t-1} - l_{t-1}I)^{-1}v_i^T\right]/\Phi_{t-1}, \tag{11}$$

where $\sum_{i=1}^n p_i = 1$. We then sample $m$ nodes into $B_m$ according to $\{p_i\}_{i=1}^n$. For each node $i \in B_m$, the corresponding weight is given by $s_i = \frac{\epsilon}{p_i \Phi_{t-1}}$, $0 < \epsilon < 1$. After obtaining the candidate set $B_m$, we apply the informative node selection criterion $\Delta_i$ introduced in Section 3.2, i.e., selecting the node $i = \arg\max_{i \in B_m} \Delta_i$, and update the query set and weights as follows:

$$\text{if } i \in \mathcal{S}_{t-1}^c : \ \mathcal{S}_t = \mathcal{S}_{t-1} \cup \{i\}, \ \mathcal{W}_t = \mathcal{W}_{t-1} \cup \{s_i\},$$

$$\text{if } i \in \mathcal{S}_{t-1} : s_i = s_i + \frac{\epsilon}{p_i \Phi_{t-1}}.$$

We then update the lower and upper eigenvalue bounds as follows:

$$u_t = u_{t-1} + \frac{\epsilon}{\Phi_{t-1}(1 - \epsilon)}, \ l_t = l_{t-1} + \frac{\epsilon}{\Phi_{t-1}(1 + \epsilon)}. \tag{12}$$

The update rule ensures that $u_t - l_t$ increases at a slower rate than $u_t$, leading to the convergence of the gap between the largest and smallest eigenvalues of $\tilde{\mathbf{X}}_\mathcal{S}^T W_S \tilde{\mathbf{X}}_\mathcal{S}$, thereby controlling the condition number. Accordingly, the covariance matrix is updated with the selected node $i$ as:

$$C_t = C_{t-1} + \frac{\epsilon}{p_i \Phi_{t-1}} v_i^T v_i. \tag{13}$$

With the covariance matrix update rule in (13), the average increment is $\mathbf{E}(C_t) - C_{t-1} = \sum_{i=1}^n p_i s_i v_i^T v_i = \frac{\epsilon}{\Phi_{t-1}} \mathbf{I}$. Intuitively, the selected node allows all eigenvalues of $C_{t-1}$ to increase at the same rate on average. This ensures that $\lambda_{\min}(\tilde{\mathbf{X}}_\mathcal{S}^T \tilde{\mathbf{X}}_\mathcal{S})$ continues to approach $\lambda_{\min}(\tilde{\mathbf{X}}^T \tilde{\mathbf{X}}) = 1$ during the selection process, thus driving the smallest eigenvalue away from zero. Additionally, the selected node remains locally informative within the candidate set $B_m$. Compared with the entire set of nodes, selecting from a subset serves as a regularization on informativeness maximization, achieving a balance between informativeness and representativeness for node queries.

### 3.4 Node query algorithm and graph signal recovery

We summarize the biased sampling selection strategy in Algorithm 1. At a high level, each step in the biased sampling query strategy consists of two stages. First, we use randomized spectral sparsification to sample $m \ll n$ nodes and collect them into a candidate set $B_m$. Intuitively, the covariance matrix on the updated $\mathcal{S}$ maintains lower-bounded eigenvalues if a node from $B_m$ is added to $\mathcal{S}$. In the second stage, we select one node from $B_m$ based on the informativeness criterion in Section 3.2 to achieve a significant frequency increase in (7).

For initialization, the dimension of the network spectrum $d$, the size of the candidate set $m$, and the constant $0 < \epsilon < 1$ for spectral sparsification need to be specified. Based on the discussion at the end of Section 3.1, the dimension of the function space $\mathbf{H}_\omega(\mathbf{X}, \mathbf{A})$ is at most $\mathcal{B}$, where $\mathcal{B}$ is the budget for label queries. Therefore, we can set $d = \min\{p, \mathcal{B}\}$. The parameters $m$ and $\epsilon$ jointly control the condition number $\frac{\lambda_{\max}(\tilde{\mathbf{X}}_\mathcal{S}^T W_S \tilde{\mathbf{X}}_\mathcal{S})}{\lambda_{\min}(\tilde{\mathbf{X}}_\mathcal{S}^T W_S \tilde{\mathbf{X}}_\mathcal{S})}$.

---

**Algorithm 1** Biased Sampling Query Algorithm

---

**Require:** $t = 0$, $C_0 = 0$, the set of query nodes $\mathcal{S}_0 = \emptyset$, the set of node weights $\mathcal{W}_0 = \emptyset$, spectral dimension $d$, size of candidate set $m$, constant $0 < \epsilon < 1/m$, query budget $\mathcal{B}$.
**Initialization**: Compute SVD decomposition $\mathbf{U}_d^T \mathbf{X} = V_1 \Sigma V_2^T$, and set $\tilde{\mathbf{X}} = \mathbf{U}_d V_1$, $r = \text{rank}(\tilde{\mathbf{X}})$, $u_0 = 2r/\epsilon$, $l_0 = -2r/\epsilon$, $\kappa = 2r(1 - m^2 \epsilon^2)/(m\epsilon^2)$.
**while** $\mathcal{B} > 0$ **do**
    **Step 1**: Calculate $\Phi_t$ as in (10) and the node-wise probabilities $\{p_i\}_{i=1}^n$ using (11).
    **Step 2**: Sample $m$ nodes with replacement according to probabilities $\{p_i\}_{i=1}^n$ to form the candidate set $B_m$.
    **Step 3**: Select node $i$ as $i = \arg\max_{i \in B_m} \Delta_i$ and calculate its weight $w_i = \frac{\epsilon}{p_i \Phi_t}$.
        **If** $i \notin \mathcal{S}_t$, then update $\mathcal{S}_{t+1} = \mathcal{S}_t \cup \{i\}$ and $\mathcal{W}_{t+1} = \mathcal{W}_t \cup \{s_i\}$ with $s_i = \frac{w_i}{\kappa}$.
        **Else if** $i \in \mathcal{S}_t$, then update $s_i = s_i + \frac{w_i}{\kappa}$.
    **Step 4**: Update $C_t$, $u_t$, $l_t$, $\mathcal{B}$ and $t$ as:

$$C_{t+1} = C_t + w_i v_i^T v_i, \quad u_{t+1} = u_t + \frac{\epsilon}{\Phi_t(1 - m\epsilon)}, \quad l_{t+1} = l_t + \frac{\epsilon}{\Phi_t(1 + m\epsilon)},$$
$$\mathcal{B} = \mathcal{B} - 1, \quad t = t + 1.$$

**end while**
**Query**: Label all nodes in $\mathcal{S}$ through an external annotator.
**Output**: Set of queried nodes $\mathcal{S}$, annotated responses $\{Y_i \mid i \in \mathcal{S}\}$, smoothed covariates $\tilde{\mathbf{X}}_\mathcal{S}$, and weights of queried nodes $\mathcal{W}$.

---

Based on the output from Algorithm 1, we solve the weighted least squares problem in (8):

$$\hat{\beta} = \arg\min_{\tilde{\beta}} \sum_{i \in \mathcal{S}} s_i \left( Y(i) - (\tilde{\mathbf{X}}_\mathcal{S})_i. \tilde{\beta} \right)^2, \tag{14}$$

and recover the graph signal on the entire network as $\hat{\mathbf{f}} = \tilde{\mathbf{X}}\hat{\beta}$. The proposed method is illustrated for the regression task, with an extension to the classification task discussed in Appendix B.2.

# 4 Theoretical Analysis

In this section, we present a theoretical analysis of the proposed node query strategy. The results are divided into two parts: the first focuses on the local information gain of the selection process, while the second examines the global performance of graph signal recovery. Given a set of query nodes $\mathcal{S}$, the information gain from querying the label of a new node $i$ is measured as the increase in bandwidth frequency, defined as $\Delta_i := \omega(\mathcal{S} \cup \{i\}) - \omega(\mathcal{S})$. We provide a step-by-step analysis of the proposed method by comparing the increase in bandwidth frequency with that of random selection.

**Theorem 4.1** *Define $d_{min} = \min_i\{d_i\}$, where $d_i$ denotes the degree of node $i$. Let $\mathcal{S}$ represent the set of queried nodes prior to the $s^{th}$ selection. Denote the adjacency matrix of the subgraph excluding $\mathcal{S}$ as $\mathbf{A}_{(n-|\mathcal{S}|)\times(n-|\mathcal{S}|)}$. Let $\Delta_s^R$ and $\Delta_s^B$ denote the increase in bandwidth frequency resulting from the $s^{th}$ label query on a node selected by random sampling and the proposed sampling method, respectively. Let $j^*$ denote the node with the largest magnitude in the eigenvector corresponding to the smallest non-zero eigenvalue of $\mathcal{L}_{\mathcal{S}^c}$. Then we have:*

$$\mathbf{E}(\Delta_s^R) = \Omega(\frac{1}{n}) \ (1), \quad and \ \ \mathbf{E}(\Delta_s^B) - \mathbf{E}(\Delta_s^R) > \Omega(\frac{1}{\eta_0 \eta_1^3 d_{min}^2}) - \Omega(\frac{1}{n}) \ (2),$$

*where $f = \Omega(g)$ if $c_1 \leq |\frac{f}{g}| \leq c_2$ for constants $c_1, c_2$ when $n$ is sufficient large. Inequality (2) holds given $m$ satisfying*

$$(\frac{n - m - d_{min}}{n - m})^m (\frac{n - m - d_{min}}{n - d_{min}})^{d_{min}} \sqrt{d_{min}} = \mathcal{O}(1).$$

*The expectation $\mathbf{E}(\cdot)$ is taken over the randomness of node selection. Both $\eta_0, \eta_1$ are network-related quantities, where $\eta_0 \triangleq \#|\{i : |\frac{d_i - d_{j^*}}{d_{min}}| \leq 1\}|$ and $\eta_1 \triangleq \max_i(\frac{d_i}{d_{min}})$.*

Theorem 4.1 provides key insights into the information gain achieved through different node label querying strategies. While random selection yields a constant average information gain, the proposed biased sampling method guarantees a higher information gain under mild assumptions.

In Theorem 4.2, we provide the generalization error bound for the proposed sampling method under the weighted OLS estimator. To formally state Theorem 4.2, we first introduce the following two assumptions:

**Assumption 1** For the underlying graph signal $\mathbf{f}$, there exists a bandwidth frequency $\omega_0$ such that $\mathbf{f} \in \mathbf{H}_{\omega_0}(\mathbf{X}, \mathbf{A})$.

**Assumption 2** The observed node-wise response $Y_i$ can be decomposed as $Y_i = \mathbf{f}(i) + \xi_i$, where $\{\xi_i\}_{i=1}^n$ are independent random variables with $\mathbf{E}(\xi_i) = 0$ and $\mathbf{Var}(\xi_i) \leq \sigma^2$.

**Theorem 4.2** *Under Assumptions 1 and 2, for the graph signal estimation $\hat{\mathbf{f}}$ obtained by training (14) on $\mathcal{B}$ labeled nodes selected by Algorithm 1, with probability greater than $1 - \frac{2m}{t}$, where $t > 2m$, we have*

$$\mathbf{E}_Y\|\hat{\mathbf{f}} - \mathbf{f}\|_2^2 \leq \mathcal{O}\Big(\frac{r_d t}{\mathcal{B}} + 2(\frac{r_d t}{\mathcal{B}})^{3/2} + (\frac{r_d t}{\mathcal{B}})^2\Big) \times (n\sigma^2 + \sum_{i>d, i\in supp(\mathbf{f})} \alpha_i^2) + \sum_{i>d, i\in supp(\mathbf{f})} \alpha_i^2, \tag{15}$$

*where $\alpha_i := \langle \mathbf{f}, U_i \rangle$, $supp(\mathbf{f}) := \{i : 1 \leq i \leq n, |\alpha_i| > 0\}$ and $r_d = rank(\mathbf{U}_d^T \mathbf{X})$. $\mathbf{E}_Y(\cdot)$ denotes the expected value with respect to the randomness in observed responses.*

Theorem 4.2 reveals the trade-off between informativeness and representativeness in graph-based active learning, which is controlled by the spectral dimension $d$. Since $r_d$ is a monotonic function of $d$, a larger $d$ reduces representativeness among queried nodes, thereby increasing variance in controlling the condition number (i.e., the first three terms). On the other hand, a larger $d$ reduces approximation bias to the true graph signal (i.e., the fifth and last terms) by including more informative nodes for capturing less smoothed signals.

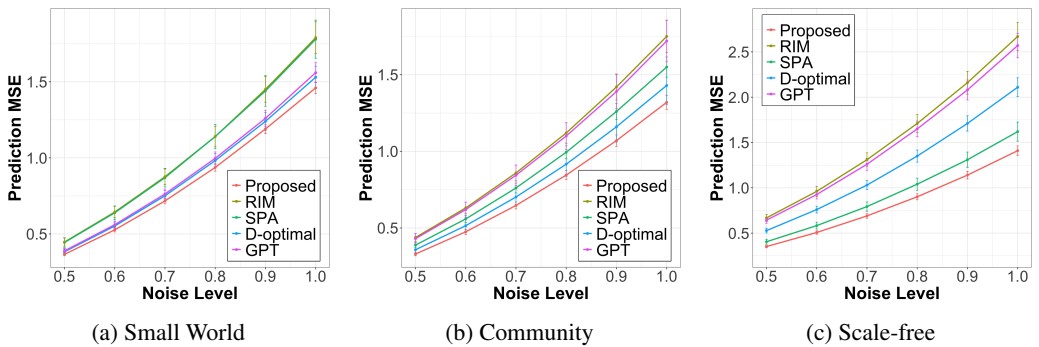

| (a) Small World | (b) Community | (c) Scale-free |

Figure 1: Prediction performance on unlabeled nodes at different levels of labeling noise ($\sigma^2$). All three simulated networks have $n = 100$ nodes, with the number of labeled nodes fixed at 25.

## 5  Numerical Studies

In this section, we conduct extensive numerical studies to evaluate the proposed active learning strategy for node-level prediction tasks on both synthetic and real-world networks. For the synthetic networks, we focus on regression tasks with continuous responses, while for the real-world networks, we consider classification tasks with discrete node labels.

### 5.1  Synthetic networks

We consider three different network topologies generated by widely studied statistical network models: the Watts–Strogatz model [33] for small-world properties, the Stochastic block model [13] for community structure, and the Barabási–Albert model [3] for scale-free properties.

Node responses are generated as $Y = \mathbf{f} + \xi$, where $\mathbf{f}$ is the true graph signal and $\xi \sim N(0, \sigma^2 I_n)$ is Gaussian noise. The true signal is constructed as $\mathbf{f} = \mathbf{U}_d \beta$, where $\beta$ is the linear coefficient and $\mathbf{U}_d$ denotes the leading $d$ eigenvectors of the normalized graph Laplacian of the synthetic network. Since our theoretical analysis assumes that the observed node covariates $\mathbf{X}$ contain noise, we generate $\mathbf{X}$ as a perturbed version of $\mathbf{U}_d$ by adding non-leading eigenvectors of the normalized graph Laplacian. The detailed simulation settings can be found in Appendix C.1.

We compare our algorithm with several offline active learning methods: 1. **D-optimal** [26] selects subset of nodes $\mathcal{S}$ to maximize determinant of observed covariate information matrix $\mathbf{X}_{\mathcal{S}}^T \mathbf{X}_{\mathcal{S}}$. 2. **RIM** [38] selects nodes to maximize the number of influenced nodes. 3. **GPT** [22] and **SPA** [27] split the graph into disjoint partitions and select informative nodes from each partition.

After the node query step, we fit the weighted linear regression from (14) on the labeled nodes, using the smoothed covariates $\tilde{\mathbf{X}}$, to estimate the linear coefficient $\hat{\beta}$ and predict the response $\hat{\mathbf{Y}}$ for the unlabeled nodes. In Figure 1, we plot the prediction MSE of the proposed method against baselines on unlabeled nodes for various levels of labeling noise $\sigma^2 \in (0.5, 0.6, 0.7, 0.8, 0.9, 1)$. The results show that the proposed method significantly outperforms all baselines across all simulation settings and exhibits strong robustness to noise. The inferior performance of the baselines can be attributed to several factors. **D-optimal** and **RIM** fail to account for noise in the node covariates. Meanwhile, partition-based methods like **GPT** and **SPA** are highly sensitive to hyperparameters, such as the optimal number of partitions, which limits their generalization to networks lacking a clear community structure.

### 5.2  Real-world networks

We evaluate the proposed method for node classification tasks on real-world datasets, which include five networks with varying homophily levels (*high to low:* **Cora**, **PubMed**, **Citeseer**, **Chameleon** and **Texas**) and two large-scale networks (**Ogbn-Arxiv** and **Co-Physics**). In addition to the offline methods described in Section 5.1, we also compare our approach with two GNN-based online active learning methods **AGE** [4] and **IGP** [39]. In each GNN iteration, **AGE** selects nodes to maximize a linear combination of heuristic metrics, while **IGP** selects nodes that maximize information gain propagation.

Table 1: Test accuracy (Micro-F1%) on five real-world networks with varying levels of homophily. The edge homophily ratio $h$ of a network is defined as the fraction of edges that connect nodes with the same class label. A higher $h$ indicates a network with stronger homophily.

| | Cora ($h = 0.81$) | | | Pubmed ($h = 0.80$) | | | Citeseer ($h = 0.74$) | | | Chameleon ($h = 0.23$) | | | Texas ($h = 0.11$) | | |
|---|---|---|---|---|---|---|---|---|---|---|---|---|---|---|---|
| #labeled nodes | 35 | 70 | 140 | 15 | 30 | 60 | 30 | 60 | 120 | 50 | 75 | 100 | 15 | 30 | 45 |
| Random | 68.2±1.3 | 74.5±1.0 | 78.9±0.9 | 71.2±1.8 | 74.9±1.6 | 78.4±0.5 | 57.7±0.8 | 65.3±1.4 | 70.7±0.7 | 22.4±2.6 | 22.1±2.5 | 21.8±2.1 | 67.0±3.3 | 69.9±3.3 | 73.8±3.2 |
| AGE | 72.1±1.1 | 78.0±0.9 | 82.5±0.5 | 74.9±1.1 | 77.5±1.2 | 79.4±0.7 | 65.3±1.1 | 67.7±0.5 | 71.4±0.5 | 30.0±4.5 | 28.2±4.9 | 28.6±5.0 | 67.9±2.6 | 68.8±3.3 | 72.1±3.6 |
| GPT | 77.4±1.6 | 81.6±1.2 | **86.5**±1.2 | 77.0±3.1 | 79.9±2.8 | 81.5±1.6 | 67.9±1.8 | 71.0±2.4 | 74.0±2.0 | 14.1±2.5 | 15.8±2.2 | 16.4±2.4 | 72.6±2.0 | 72.5±3.6 | 74.6±1.8 |
| RIM | 77.5±0.8 | 81.6±1.1 | 84.1±0.8 | 75.0±1.5 | 77.2±0.6 | 80.2±0.4 | 67.5±0.7 | 70.0±0.6 | 73.2±0.7 | **35.5**±3.7 | **42.8**±3.0 | 34.4±3.5 | 68.5±3.7 | 78.4±3.0 | 74.6±3.7 |
| IGP | 77.4±1.7 | 81.7±1.6 | 86.3±0.7 | 78.5±1.2 | **82.3**±1.4 | **83.5**±0.5 | 68.2±1.1 | 72.1±0.9 | **75.8**±0.4 | 32.5±3.6 | 33.7±3.1 | 33.4±3.5 | 70.8±3.7 | 69.9±3.3 | 76.1±3.6 |
| SPA | 76.5±1.9 | 80.3±1.6 | 85.2±0.6 | 75.4±1.6 | 78.3±2.0 | 73.5±1.2 | 66.4±2.2 | 69.3±1.7 | 73.5±2.0 | 30.2±3.2 | 28.5±2.9 | 31.0±4.4 | 72.0±3.2 | 72.5±3.1 | 74.6±2.1 |
| Proposed | **78.4**±1.7 | **81.8**±1.8 | **86.5**±1.1 | **78.9**±1.1 | 79.1±0.6 | 82.3±0.6 | **69.1**±1.0 | **72.2**±1.3 | 75.5±0.8 | 35.1±2.8 | 35.7±3.0 | **37.2**±3.0 | **75.0**±1.9 | **79.5**±0.8 | **80.4**±2.7 |

Table 2: Test accuracy (Macro-F1% and Micro-F1%) on two real-world large-scale networks: Ogbn-Arxiv ($n = 169,343$) and Co-Physics ($n = 34,493$).

| | Ogbn-Arxiv (Macro-F1) | | | | Ogbn-Arxiv (Micro-F1) | | | | Co-Physics (Macro-F1) | | |
|---|---|---|---|---|---|---|---|---|---|---|---|
| #labeled nodes | 160 | 320 | 640 | 1280 | 160 | 320 | 640 | 1280 | 10 | 20 | 40 |
| Random | 21.9±1.4 | 27.6±1.5 | 33.0±1.4 | 37.2±1.1 | 52.3±0.8 | 56.4±0.8 | 60.0±0.7 | 63.5±0.4 | 58.3±13.8 | 66.9±10.1 | 78.3±7.1 |
| AGE | 20.4±0.9 | 25.9±1.1 | 31.7±0.8 | 36.4±0.8 | 48.3±2.3 | 54.9±1.6 | 60.0±0.7 | 63.5±0.3 | 63.7±7.8 | 71.0±8.8 | 82.4±3.9 |
| GPT | 24.2±0.7 | 29.5±0.8 | 36.4±0.5 | 41.0±0.5 | 52.3±0.9 | 56.8±0.8 | 60.7±0.6 | 63.6±0.5 | 75.8±2.7 | 85.8±0.3 | 88.9±0.3 |
| Proposed | **25.8**±1.3 | **34.3**±1.4 | **38.3**±1.2 | **41.3**±1.3 | **53.1**±1.3 | **58.0**±1.0 | **62.3**±1.6 | **64.8**±1.0 | **83.5**±0.8 | **86.8**±1.3 | **89.2**±1.2 |

Table 3: Average query time (in seconds) per node.

| Dataset | Size | Time |
|---|---|---|
| Texas | 183 | 0.19 ± 0.03 |
| Chameleon | 2,277 | 0.34 ± 0.18 |
| Cora | 2,708 | 0.30 ± 0.19 |
| Citeseer | 3,327 | 0.26 ± 0.07 |
| Pubmed | 19,717 | 0.48 ± 0.25 |
| Co-Physics | 34,493 | 1.08 ± 0.43 |
| Ogbn-Arxiv | 169,343 | 2.11 ± 0.33 |

Unlike regression, node classification with GNNs is a widely studied area of research. Previous works [22, 30, 38, 39] have demonstrated that the prediction performance of various active learning strategies on unlabeled nodes remains relatively consistent across different types of GNNs. Therefore, we employ Simplified Graph Convolution (SGC) [34] as the GNN classifier due to its straightforward theoretical intuition. Since SGC is essentially multi-class logistic regression on low-pass-filtered covariates, it can be approximately viewed as a special case of the regression model defined in (14). Thus, we conjecture that our theoretical analysis can also be extended to classification tasks and leave its formal verification for future work.

The results in Figure 1 demonstrate that the proposed algorithm is highly competitive with baselines across real-world networks with varying degrees of homophily. Our method achieves the best performance on Cora (highest homophily) and Texas (lowest homophily, i.e., highest heterophily) and is particularly effective when the labeling budget is most limited. To handle heterophily in networks like Chameleon and Texas, we expand the graph signal subspace $\mathbf{U}_d$ in Algorithm 1 to $\mathbf{U}_d = \{U_1, \cdots, U_d, U_{n-d+1}, \cdots, U_n\}$, combining eigenvectors corresponding to the $d$ smallest and $d$ largest eigenvalues. Admittedly, relying on a priori knowledge of label construction may be unrealistic, so developing adaptive methods for designing the signal subspace to effectively handle both homophily and heterophily remains a promising direction for future research.

Table 2 summarizes the performance on two large-scale networks. The greatest improvement is observed in the Macro-F1 score on Ogbn-Arxiv, with an increase of up to 4.8% at 320 labeled nodes. Moreover, Table 3 demonstrates that our algorithm scales efficiently to large networks, with the time cost of querying a single node being approximately 2 seconds when $n = 169,343$.

## 6 Conclusion

We propose a graph-based offline active learning framework for node-level tasks. Our node query strategy effectively leverages both the network structure and node covariate information, demonstrating robustness to diverse network topologies and node-level noise. We provide theoretical guarantees for controlling generalization error, uncovering a novel trade-off between informativeness and representativeness in active learning on graphs. Empirical results demonstrate that our method performs strongly on both synthetic and real-world networks, achieving competitiveness with state-of-the-art methods on benchmark datasets. Future work could explore extensions to an online active learning setting that iteratively incorporates node response information to further enhance query efficiency. Additionally, scalability on large graphs could be improved by utilizing the Lanczos method [1] or Chebyshev polynomial approximation [16] during node selection.

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

# A Proofs

## A.1 Proof of Theorem 3.1

*Proof*: Consider the threshold frequency $\omega$ defined as

$$\omega < \omega(\mathcal{S}) := \inf_{\mathbf{g} \in \text{Proj}_{\mathbf{L}_{\mathcal{S}^c}} \text{span}(\mathbf{X})} \omega_{\mathbf{g}}, \tag{16}$$

and notice that 16 is true if and only if

$$\text{Proj}_{\mathbf{L}_\omega} \text{span}(\mathbf{X}) \cap \text{Proj}_{\mathbf{L}_{\mathcal{S}^c}} \text{span}(\mathbf{X}) = \{0\}. \tag{17}$$

$\Rightarrow$ For $\forall \phi \in \text{Proj}_{\mathbf{L}_\omega} \text{span}(\mathbf{X}) \cap \text{Proj}_{\mathbf{L}_{\mathcal{S}^c}} \text{span}(\mathbf{X})$ and $\forall \mathbf{f} \in \text{Proj}_{\mathbf{L}_\omega} \text{span}(\mathbf{X})$, we have $\mathbf{g} = \phi + \mathbf{f} \in \text{Proj}_{\mathbf{L}_\omega} \text{span}(\mathbf{X})$ by closure under addition and $\mathbf{f}(S) = \mathbf{g}(S)$. Since $\text{Proj}_{\mathbf{L}_\omega} \text{span}(\mathbf{X})$ can be identified by $\mathcal{S}$, one must have $\mathbf{f} = \mathbf{g}$, which implies $\phi = 0$. Hence, 17 is true.

$\Leftarrow$ Given 17 is true, assume that $\text{Proj}_{\mathbf{L}_\omega} \text{span}(\mathbf{X})$ cannot be identified by $\mathcal{S}$. Then, by definition, there exists $\mathbf{f}_1, \mathbf{f}_2 \in \text{Proj}_{\mathbf{L}_\omega} \text{span}(\mathbf{X})$ such that $\mathbf{f}_1(S) = \mathbf{f}_2(S)$ and $\mathbf{f}_1 \neq \mathbf{f}_2$. Since $(\mathbf{f}_1 - \mathbf{f}_2)(S) = 0$, $\mathbf{f}_1 - \mathbf{f}_2 \in \text{Proj}_{\mathbf{L}_{\mathcal{S}^c}} \text{span}(\mathbf{X})$. By clousure under addition, $\mathbf{f}_1 - \mathbf{f}_2 \in \text{Proj}_{\mathbf{L}_\omega} \text{span}(\mathbf{X})$. However, $\mathbf{f}_1 - \mathbf{f}_2 \in \text{Proj}_{\mathbf{L}_\omega} \text{span}(\mathbf{X}) \cap \text{Proj}_{\mathbf{L}_{\mathcal{S}^c}} \text{span}(\mathbf{X})$ is a contradiction with 17. Therefore, $\text{Proj}_{\mathbf{L}_\omega} \text{span}(\mathbf{X})$ can be identified by $\mathcal{S}$.

## A.2 Proof of Theorem 4.1

*Proof*: Let $\mathbf{U} = \{U_1, U_2, \cdots, U_n\}$ be the eigenvectors of $\mathcal{L} = \mathbf{I} - \mathbf{D}^{-1/2}\mathbf{A}\mathbf{D}^{-1/2}$ and $\Lambda = \text{diag}(\lambda_1, \lambda_2, \cdots, \lambda_n)$. Without loss of generality, we analyze the case when $k = 1$ and the case of no repeated eigenvalues. The analysis for the case of repeated eigenvalues can be similar performed with matrix perturbation analysis on degenerate case [2].

After selecting node $i$, we have the reduced Laplacian matrix $\mathcal{L}^*$

$$\mathcal{L}^* = \mathcal{L} + \mathcal{L}^{(-i)}, \quad \text{where} \quad \mathcal{L}^{(-i)} = - \begin{pmatrix} 0 & \cdots & \mathcal{L}_{1i} & \cdots & 0 \\ \vdots & \ddots & \vdots & \vdots & \vdots \\ \mathcal{L}_{i1} & \cdots & \mathcal{L}_{ii} & \cdots & \mathcal{L}_{in} \\ \vdots & \vdots & \vdots & \ddots & \vdots \\ 0 & \cdots & \mathcal{L}_{ni} & \cdots & 0 \end{pmatrix}$$

Define $\Lambda^*$ as the diagonal matrix containing eigenvalues of $\mathcal{L}^*$. Using the first order perturbation analysis [2], we have $\Lambda^* \approx \Lambda + \text{diag}(\mathbf{U}^T \mathcal{L}^{(-i)} \mathbf{U})$. Let $\mathcal{L}_{\cdot i}$ and $\mathcal{L}_{i\cdot}$ be the $i^{th}$ column and row of $\mathcal{L}$, respectively. Since

$$\mathcal{L}^{(-i)} U_j = \begin{pmatrix} -\mathcal{L}_{1i}\mathbf{U}_{ij} \\ \vdots \\ -\mathcal{L}_{i\cdot}^{(-i)}U_j \\ \vdots \\ -\mathcal{L}_{ni}\mathbf{U}_{ij} \end{pmatrix} = -\mathcal{L}_{\cdot i}\mathbf{U}_{ij} + \begin{pmatrix} 0 \\ \vdots \\ \mathcal{L}_{ii}\mathbf{U}_{ij} - \mathcal{L}_{i\cdot}^{(-i)}U_j \\ \vdots \\ 0 \end{pmatrix} = -\mathcal{L}_{\cdot i}\mathbf{U}_{ij} + \begin{pmatrix} 0 \\ \vdots \\ (\mathcal{L}_{ii} - \lambda_j)\mathbf{U}_{ij} \\ \vdots \\ 0 \end{pmatrix}$$

we have

$$\begin{aligned}
U_j^T \mathcal{L}^{(-i)} U_j &= -\mathbf{U}_{ij}(\mathcal{L}_i.U_j) + (\mathcal{L}_{ii} - \lambda_j)\mathbf{U}_{ij}^2 \\
&= -\mathbf{U}_{ij}(\lambda_j \mathbf{U}_{ij}) + (\mathcal{L}_{ii} - \lambda_j)\mathbf{U}_{ij}^2 \\
&= (1 - 2\lambda_j)\mathbf{U}_{ij}^2 \text{ since the diagonal entry of } \mathcal{L} \text{ is } 1
\end{aligned}$$

Therefore,

$$\Lambda^* \approx \Lambda + \begin{pmatrix} (1 - 2\lambda_1)\mathbf{U}_{i1}^2 & & \\ & \ddots & \\ & & (1 - 2\lambda_n)\mathbf{U}_{in}^2 \end{pmatrix}$$

Assume $U_1$ corresponding to the smallest non-zero eigenvalue of $\Lambda^*$, then the increase of bandwidth frequency for node $i$ is $\mathbf{U}_{i1}^2$.

For random selection, where node $i$ is queried with uniform probability,

$$\mathbf{E}(\Delta_{\mathrm{R}}) = \frac{1}{n}\sum_{i=1}^{n} \mathbf{U}_{i1}^2 \propto \frac{1}{n}$$

For the proposed selection method, define $\mathbf{R} = \mathrm{diag}(r_1, r_2, \cdots, r_n)$, where $r_i = \frac{d_i}{d_{min}}$, then

$$\begin{aligned}
\mathcal{L} &= \mathbf{I} - \mathbf{D}^{-1/2}\mathbf{A}\mathbf{D}^{-1/2} \\
&= \underbrace{\mathbf{I} + \epsilon'\mathbf{D}}_{\mathbf{A_0}} + \epsilon'\left(\underbrace{(\epsilon' d_{min})^{-1}\mathbf{R}^{-\frac{1}{2}}\mathbf{A}\mathbf{R}^{-\frac{1}{2}} - \mathbf{D}}_{\mathbf{A_1}}\right) \\
&= \mathbf{A_0} + \epsilon'\mathbf{A_1}
\end{aligned}$$

Next we perform matrix perturbation analysis, define

$$U_0 = I,\ \Lambda_0 = A_0,\ \Lambda = \Lambda_0 + \epsilon'\Lambda_1,\ U = U_0 + \epsilon'U_1$$

where $\Lambda, U$ approximate eigenvalues and eigenvectors of $\Lambda^*$. Denote $(U_1)_k$ as the $k^{th}$ column of $U_1$ and assume $d_i \neq d_j$, we have

$$(U_1)_k = \sum_{i \neq k} \frac{(U_0)_i^\top A_1 (U_0)_k}{(\Lambda_0)_k - (\Lambda_0)_i}(U_0)_i = \sum_{i \neq k} \frac{A_{ik}}{(\varepsilon')^2 (d_i - d_k)\sqrt{d_i}\sqrt{d_k}}e_i$$

$$(U)_k = (U_0)_k + \varepsilon'(U_1)_k$$

To satisfy $\|U_k\|_2 = 1$, we multiply $\tau$ as

$$(\tau U)_k = (\tau U_0)_k + \varepsilon'(\tau U_1)_k$$

$\Rightarrow$ recalculate $U_1$ by $\tau U_0$ as $(U_1)_k = \sum_{i \neq k} \frac{A_{ik}\tau^3}{(\varepsilon')^2 (d_i - d_k)\sqrt{d_i}\sqrt{d_k}}e_i$

$\Rightarrow (\tau U)_k = \tau e_k + \sum_{i \neq k} \frac{A_{ik}\tau^4}{\varepsilon'(d_i - d_k)\sqrt{d_i}\sqrt{d_k}}e_i$, choose $\varepsilon' = \tau^3 \times \frac{1}{\sqrt{d_k}d_{\min}^{\frac{3}{2}}}$

$\Rightarrow$ then $\|\tau U_k\|_2 = 1$ if $\tau = \dfrac{1}{\sqrt{1 + \sum_{i \neq k} \frac{A_{ik}}{\left(\frac{d_i - d_k}{d_{\min}}\right)^2 \frac{d_i}{d_{\min}}}}}$

Then we consider the normalized $\tau U$ as $U$ in the following analysis. Assume $(U)_k$ is the eigenvector to the smallest non-zero eigenvalue, then at the $t-1$ step

$$\mathbf{E}(\Delta) = \mathbf{E}(\mathbf{U}_{ik}^2)$$

where $\mathbf{E}$ is in terms of the randomness in the proposed sampling procedure.

Based on the approximation

$$(U)_k = \tau e_k + \sum_{i \neq k} \frac{A_{ik}\tau}{\frac{d_i - d_k}{d_{\min}} \times \sqrt{\frac{d_i}{d_{\min}}}}$$

then

$$\#\left|\{\mathbf{U}_{ik} \neq 0\}_{i=1}^n\right| = 1 + d_k > d_{\min}$$

Define $S = \{i \in \{1, 2, \cdots, n\} : \mathbf{U}_{ik} \neq 0\}$ and $\mathbf{P_1} = P(\text{the node being selected to query} \notin S)$. Take $p = \min_{i \in S} \mathbf{P}(i), q = \max_{i \notin S} \mathbf{P}(i)$ and $\mathbf{P}(\cdot)$ denotes the probability of being selected into candidate set size $m$. Denote $k = d_{min}$, we first upper bound $\mathbf{P_1}$ as

$$\mathbf{P_1} \leq \frac{\binom{n-k}{m} q^m (1-q)^{n-k-m}(1-p)^k}{\sum_{i=0}^k \binom{n-k}{m-i} q^{m-k}(1-q)^{n-k-(m-i)}\binom{k}{i} p^i (1-p)^{k-i}}$$

$$= \frac{\binom{n-k}{m}}{\sum_{i=0}^k \binom{n-k}{m-i}\binom{k}{i}\eta^i}, \text{ where } \eta = \frac{p(1-q)}{(1-p)q}$$

We calculate the denominator as

$$\sum_{i=0}^k \binom{n-k}{m-i}\binom{k}{i}\eta^i = c_0 \sqrt{\sum_{i=0}^k \binom{n-k}{m-i}^2 \binom{k}{i}^2} \sqrt{\sum_{i=0}^k \eta^{2i}}$$

$$\geq \frac{c_0}{\sqrt{k+1}}\binom{n}{m}\sqrt{\frac{1-\eta^{2k+2}}{1-\eta^2}} > \frac{c_0}{\sqrt{k+1}}\binom{n}{m}.$$

In addition, by Stirling's approximation, when $n$ is large

$$\binom{n}{m} \sim \sqrt{\frac{n}{2\pi m(n-m)}} \cdot \frac{n^n}{m^m(n-m)^{n-m}}$$

$$\binom{n-k}{m} \sim \sqrt{\frac{n-k}{2\pi m(n-k-m)}} \cdot \frac{(n-k)^{n-k}}{m^m(n-k-m)^{n-k-m}}$$

then combining the above simplification, we have

$$\mathbf{P_1} < \sqrt{\frac{n-k}{n} \cdot \frac{n-m}{n-k-m}} \cdot \frac{(n-k)^{n-k}}{n^n} \cdot \frac{(n-m)^{n-m}}{(n-k-m)^{n-k-m}} \times \frac{\sqrt{k+1}}{c_0}$$

$$\leq \left(\frac{n-k-m}{n-m}\right)^m \left(\frac{n-k-m}{n-k}\right)^k \times \frac{\sqrt{k+1}}{c_0}$$

then we can lower bound the expected value of information gain as

$$\mathbf{E}(\mathbf{U}_{ik}^2) = \sum_{i \in S} \mathbf{P}(i)\mathbf{U}_{i1}^2 \geq (1-\mathbf{P_1})\min_{i \in S}\left(\mathbf{U}_{ik}^2\right)$$

Notice that $\mathbf{P_1}$ is a monotone decreasing function of $m$ given $n$ and $k$ fixed, then we can select a $m_0$ such that

$$\left(\frac{n-k-m_0}{n-m_0}\right)^{m_0}\left(\frac{n-k-m_0}{n-k}\right)^k \sqrt{k} \leq \frac{\delta}{c_0},$$

where $\delta < 1$ is a constant, therefore $\mathbf{E}(\mathbf{U}_{ik}^2) > (1-\delta)\min_{i \in S}\left(\mathbf{U}_{ik}^2\right)$.

Next we lower bound the quantity $\min_{i \in S} \left( \mathbf{U}_{ik}^2 \right)$. Denote $\eta_1 := \max_i \left( \frac{d_i}{d_{min}} \right)$ and $\eta_0 := \#|\{i : |\frac{d_i - d_k}{d_{min}}| \leq 1\}|$, we calculate the lower bound for $\min_{i \in S} \left( \mathbf{U}_{ik}^2 \right)$ as

$$\min_{i \in S} \left( \mathbf{U}_{ik}^2 \right) \geq \min \left( \tau, \frac{\tau^2}{\frac{(d_i - d_k)^2}{d_{min}^2} \times \frac{d_i}{d_{min}}} \right) \quad \forall i \in S$$

$$\geq \frac{1}{\frac{(d_i - d_k)^2}{d_{min}^2} \frac{d_i}{d_{min}} + 1 + \frac{(d_i - d_k)^2}{d_{min}^2} \frac{d_i}{d_{min}} \sum_{j \neq 1, j \neq i} \frac{1}{\frac{(d_j - d_i)^2}{d_{min}^2} \frac{d_j}{d_{min}}}}$$

$$\geq \frac{1}{\frac{(d_i - d_k)^2}{d_{min}^2} \frac{d_i}{d_{min}} \left( 1 + o_n(1) + \sum_{j \neq 1, j \neq i} \frac{1}{\frac{(d_j - d_i)^2}{d_{min}^2} \frac{d_j}{d_{min}}} \right)}$$

$$\approx \frac{1}{\eta_1^3 \left( 1 + \sum_{j \in \{i : |\frac{d_i - d_k}{d_{min}}| \leq 1\}} d_{min}^2 + \sum_{j \notin \{i : |\frac{d_i - d_k}{d_{min}}| < 1\}} 1 \right)}$$

$$\geq \frac{1}{\eta_1^3 (\eta_0 d_{min}^2 + d_{min} - \eta_0)}$$

which implies

$$\min_{i \in S} \left( \mathbf{U}_{ik}^2 \right) \geq \frac{1}{\eta_1^3 (\eta_0 d_{min}^2 + d_{min} - \eta_0)}$$

As a result, as long as $m \geq m_0$ we have

$$\mathbf{E}(\mathbf{U}_{ik}^2) \geq \frac{1 - \delta}{\eta_1^3 (\eta_0 d_{min}^2 + d_{min} - \eta_0)}.$$

### A.3   Proof of Theorem 4.2

*Proof*: Based on the assumption that $\mathbf{f} \in \text{Proj}_{\mathbf{L}_{\omega_0}} \text{span}(\mathbf{X})$, we denote $d_0 = |\{1 \leq j \leq n \mid \lambda_j \leq \omega_0\}|$ and $\mathbf{U}_{d_0} = (U_1, U_2, \cdots, U_{d_0})$. Therefore, we can represent $\mathbf{f} = \mathbf{U}_{d_0} \mathbf{U}_{d_0}^T \mathbf{X} \beta$ for some parameter $\beta \in \mathbb{R}^{p \times 1}$ and $\langle \mathbf{f}, U_i \rangle = U_i^T \mathbf{X} \beta$. For the query set $S$ and the corresponding bandwidth frequency $\omega \leq \omega_0$, we similarly denote $d = |\{1 \leq j \leq n \mid \lambda_j \leq \omega\}| \leq d_0$ and $\mathbf{U}_d = (U_1, U_2, \cdots, U_d)$. We denote $\mathbf{V}_{n \times r_d} = \mathbf{U}_d V_1$ as the bases of $\text{Proj}_{\mathbf{L}_\omega} \text{span}(\mathbf{X})$ where $V_1$ is obtained from SVD decomposition $\mathbf{U}_d^T \mathbf{X} = V_1 \Sigma V_2^T$ where $(V_1)_{d \times r_d}$ and $(V_2)_{p \times r_d}$ are left and right singular vectors, respectively. The diagonal matrix $\Sigma_{r_d \times r_d}$ contains $r_d$ positive singular values with $r_d \leq \min\{d, p\}$. The estimation (11) at the end of section 3 is equivalent to the weighted regression problem of $\{(\mathbf{V}_{i\cdot}, Y_i, s_i)\}_{i \in S}$,

$$\tilde{f} = \underset{\tilde{f} \in \text{Proj}_{\mathbf{L}_\omega} \text{span}(\mathbf{X})}{\arg \min} \sum_{i \in S} s_i |Y_i - \tilde{f}(i)|^2$$

$$\Rightarrow \underset{\alpha \in \mathbb{R}^{r_d \times 1}}{\arg \min} \sum_{i=1}^{\mathcal{B}} |\sqrt{s_i} Y_i - (\sqrt{s_i} \mathbf{V}_1(i), \ldots, \sqrt{s_i} \mathbf{V}_{r_d}(i)) \alpha|^2,$$

where $|S| = \mathcal{B}$. We have the least squares solution

$$\alpha(\tilde{f}) = \left( A^\top A \right)^{-1} A^\top W Y_{\mathcal{B}} \; \textcircled{1}$$

where

$$A = \begin{pmatrix} \sqrt{s_1}\mathbf{V}_1(1) & \cdots & \sqrt{s_1}\mathbf{V}_{r_d}(1) \\ \vdots & \ddots & \vdots \\ \sqrt{s_\mathcal{B}}\mathbf{V}_1(\mathcal{B}) & \cdots & \sqrt{s_{r_d}}\mathbf{V}_{r_d}(\mathcal{B}) \end{pmatrix} \quad \text{and} \quad W = \text{diag}(\sqrt{s_1, \ldots, s_\mathcal{B}}) \qquad (18)$$

We assume $Y = \mathbf{f} + \varepsilon$, where $E(\varepsilon) = 0$ and $Var(\varepsilon) = \sigma^2$. Notice the oracle $\mathbf{f}$ satisfies

$$\mathbf{f} = \underset{f \in \text{Proj}_{\mathbf{L}_{\omega_0}} \text{span}(\mathbf{X})}{\arg\min} \sum_{i=1}^{n} \mathbf{E}_Y \left( Y_i - f^2(i) \right)$$

We decompose the space $\text{Proj}_{\mathbf{L}_{\omega_0}} \text{span}(\mathbf{X})$ as

$$\text{Proj}_{\mathbf{L}_{\omega_0}} \text{span}(\mathbf{X}) = \text{Proj}_{\mathbf{L}_{\omega}} \text{span}(\mathbf{X}) \bigoplus \left( \text{Proj}_{\mathbf{L}_{\omega}} \text{span}(\mathbf{X}) \right)^c$$

Then we decompose $\mathbf{f} = \mathbf{f}_1 + \mathbf{f}_2$, where $\mathbf{f}_1 \in \text{Proj}_{\mathbf{L}_{\omega}} \text{span}(\mathbf{X})$, $\mathbf{f}_2 \in \left( \text{Proj}_{\mathbf{L}_{\omega}} \text{span}(\mathbf{X}) \right)^c$, then

$$\mathbf{f}_1 = \underset{f \in \text{Proj}_{\mathbf{L}_{\omega}} \{\text{span}(\mathbf{X})\}}{\arg\min} \sum_{i=1}^{n} \mathbf{E}_Y \left( Y_i - f(i) \right)^2$$

Then we can represent $\mathbf{f}_1(i) = (\mathbf{V}_1(i), \ldots, \mathbf{V}_{r_d}(i))\alpha(\mathbf{f}_1)$, by solving $A\alpha(\mathbf{f}_1) = W\mathbf{f}_1$, we have

$$\alpha(\mathbf{f}_1) = \left( A^\top A \right)^{-1} A^\top W \mathbf{f}_1 \;\textcircled{2}$$

From $\textcircled{1}$ and $\textcircled{2}$, we have

$$\sum_{i=1}^{n} \left| \tilde{f}(i) - \mathbf{f}(i) \right|^2 = \|\tilde{f} - \mathbf{f}\|_2^2 \leq \|\tilde{f} - \mathbf{f}_1\|_2^2 + \|\mathbf{f}_1 - \mathbf{f}\|_2^2$$

$$\leq \left\| \alpha\left( \hat{f} \right) - \alpha(\mathbf{f}_1) \right\|_2^2 + \|\mathbf{f}_1 - \mathbf{f}\|_2^2$$

$$\leq \left\| \left( A^\top A \right)^{-1} A^\top W \left( Y_\mathcal{B} - (\mathbf{f}_1)_\mathcal{B} \right) \right\|_2^2 + \|\mathbf{f}_1 - \mathbf{f}\|_2^2$$

$$\leq \lambda_{max} \left( A^\top A \right)^{-1} \| A^\top W \left( Y_\mathcal{B} - (\mathbf{f}_1)_\mathcal{B} \right) \|_2^2 + \|\mathbf{f}_1 - \mathbf{f}\|_2^2$$

Denote $g_i = Y_i - \mathbf{f}_1(i)$, we have

$$\mathbf{E}_\mathcal{S}\|A^\top W g\|^2 = \sum_{i=1}^{r_d} \mathbf{E}_\mathcal{S} \left[ \sum_{j=1}^{\mathcal{B}} s_j^2 \mathbf{V}_i^2(j) |g_j|^2 \right]$$

Denote $\alpha_i = s_i * p_i$ for $j \in \mathcal{S}$ where $p_j$ is the probability of node $j$ being selected to query. Since

$$\mathbf{E}_\mathcal{S} \left[ s_j \mathbf{V}_i(j)(Y_j - \mathbf{f}_1(j)) \right] = \alpha_j \mathbf{E}_\mathcal{S} \left[ \frac{1}{p_j} \mathbf{V}_i(j)(Y_j - \mathbf{f}_1(j)) \right]$$

$$= \alpha_j \sum_{l=1}^{n} \mathbf{E}_Y \left[ \mathbf{V}_i(l)(Y_l - \mathbf{f}_1(l)) \right]$$

$$= 0 \;\text{ since } \mathbf{V}_i \perp \mathbf{f}_2$$

we have

$$\mathbf{E}_\mathcal{S}\|A^\top W g\| = \sum_{j=1}^{\mathcal{B}} \mathbf{E}_\mathcal{S} \left[ \sum_{i=1}^{r_d} s_j^2 \mathbf{V}_i^2(j)|g_j|^2 \right] \leq \sup_j \left( s_j \sum_{i=1}^{r_d} |\mathbf{V}_i(j)|^2 \right) \times \sum_{j=1}^{\mathcal{B}} \mathbf{E}_\mathcal{S} \left( s_j g_j^2 \right)$$

$$= \sup_j \left( s_j \sum_{i=1}^{r_d} |\mathbf{V}_i(j)|^2 \right) \times \sum_{j=1}^{\mathcal{B}} \alpha_j \mathbf{E}_\mathcal{S} \left( \frac{1}{p_j} g_j^2 \right)$$

$$= \sup_j \left( s_j \sum_{i=1}^{r_d} |\mathbf{V}_i(j)|^2 \right) \times \sum_{j=1}^{\mathcal{B}} \alpha_j \times \sum_{l=1}^{n} \mathbf{E}_Y \left( Y_l - \mathbf{f}_1(l) \right)^2$$

Notice that

$$\sum_{l=1}^{n} \mathbf{E}_Y (Y_l - \mathbf{f}_1(l))^2 = \sum_{l=1}^{n} \mathbf{E}_Y (Y_l - \mathbf{f}(l))^2 + \sum_{l=1}^{n} (\mathbf{f}(l) - \mathbf{f}_1(l))^2 = n\sigma^2 + \|\mathbf{f} - \mathbf{f}_1\|_2^2$$

Notice that $\mathbf{f} = \mathbf{U}_{d_0} \mathbf{U}_{d_0}^T \mathbf{X}\beta$ and $\mathbf{f}_1 = \mathbf{U}_d \mathbf{U}_d^T \mathbf{f}$, then $\mathbf{f} - \mathbf{f}_1 = \mathbf{U}_{d'} \mathbf{U}_{d'}^T \mathbf{X}\beta = \sum_{i>d}\langle\mathbf{f}, U_i\rangle U_i$ where $\mathbf{U}_{d'} = (U_{d+1}, \cdots, U_{d_0})$. Therefore, $\|\mathbf{f} - \mathbf{f}_1\| = \sum_{i>d, i\in\mathrm{supp}(\mathbf{f})}\langle\mathbf{f}, U_i\rangle^2$. We first state and then prove the following Lemma 1.

**Lemma A.1** *For the output from Algorithm 1 with $\mathcal{B}$ query budgets, we have*

$$\sum_{i=1}^{\mathcal{B}} \alpha_i \leq \frac{4}{3}, \quad \sup_{j\in\mathcal{S}} \left( s_j \sum_{i=1}^{r_d} |\mathbf{V}_i(j)|^2 \right) \leq 10\delta, \text{ and}$$

$$\lambda(A^\top A) \in \left[ \frac{1}{2} \times \frac{1}{\left(1 + \frac{m\sqrt{\delta}}{C_0}\right)^2}, \frac{8}{3} \times \frac{1}{1 - \left(\frac{m\sqrt{\delta}}{C_0}\right)^2} \right] \text{ with probability } 1 - \frac{2}{C},$$

*where $\delta = \frac{r_d C C_0^2}{m\mathcal{B}}$, and $C_0$ is a constant such that $C_0^2 = (m)^2 + \max(2, \frac{16}{d})m$.*

Using Lemma 1 we have

$$\mathbf{E}\|\tilde{f} - \mathbf{f}\|_2^2 \leq \lambda_{min}(A^\top A) \times \left( \sum_{j=1}^{\mathcal{B}} \alpha_j \right) \times \sup_{j\in\mathcal{S}} \left( s_j \sum_{i=1}^{r_d} |\mathbf{V}_i(j)|^2 \right) \times \sum_{j=1}^{n} \mathbf{E}_Y (Y_j - \mathbf{f}_1(j))^2 + \|\mathbf{f} - \mathbf{f}_1\|_2^2$$

$$\leq 2(1 + \frac{m\sqrt{\delta}}{C_0})^2 \times \frac{4}{3} \times 10 \times \frac{C C_0^2 r_d}{m} \times \frac{1}{\mathcal{B}} \times \sum_{j=1}^{n} \mathbf{E}_Y (Y_j - \mathbf{f}_1(j))^2 + \|\mathbf{f} - \mathbf{f}_1\|_2^2$$

$$\leq O\left( 2(\frac{r_d t}{\mathcal{B}}) + \frac{r_d t}{\mathcal{B}})^{3/2} + (\frac{r_d t}{\mathcal{B}})^2 \right) \times (n\sigma^2 + \sum_{i>d, i\in\mathrm{supp}(\mathbf{f})} \langle U_i, \mathbf{f}\rangle^2) + \sum_{i>d, i\in\mathrm{supp}(\mathbf{f})} \langle U_i, \mathbf{f}\rangle^2,$$

with probability larger than $1 - \frac{2m}{t}$ where $t > 2m$.

In the following, we prove Lemma 1 which is based on Theorem 5.2 in [5] and Lemma 3.5 and 3.6 in [17].

In the following, we denote the accumulated covariance matrix in the $j$ selection as $A_j$, the potential function as $\Phi_{u_j, l_j}(A_j) = \mathrm{Tr}[(u_j I - A_j)^{-1}] + \mathrm{Tr}[(A_j - l_j I)^{-1}]$, and $R_i(u, l, A) = v_i(uI - A)^{-1} v_i^T + v_i(A - lI)^{-1} v_i^T$, where $v_i$ is the $i$th row of $\mathbf{V}$. Notice that $\sum_{i=1}^{n} R_i = \Phi_{u,l}(A)$. At each iteration of algorithm 1, the $i$th node is selected as one of $m$ candidates with $p*_i = \frac{R_i}{\Phi}$. For the $m$ candidates, we define the following probability

$$q_i = \begin{cases} 1 - \eta & \text{if } i \text{ has maximum } \Delta_i \text{ among m candidates} \\ \frac{\eta}{m-1} & \text{otherwise} \end{cases}$$

where $0 < \eta < 1$. Notice that when $\eta$ goes to 0, the $q_i$ approximate the step 3 in Algorithm 1. Therefore, the probability of node k being query is

$$p_k = P(\text{select k}) = P(\underbrace{\text{select k}}_{Q_k} | k \text{ in } B_m) \cdot P(k \text{ in } B_m) = p_k^* \times q_k$$

$$\mathbf{E}\left(\frac{1}{p_k}v_k v_k^\top\right) = \mathbf{E}_{B_m}\mathbf{E}_{Q_k|B_m}\left(\frac{1}{P(\text{select k})}v_k v_k^\top\right)$$

$$= \mathbf{E}_{B_m}\left(\sum_{k\in B_m} P(\text{select } k|k\in B_m)\cdot\frac{1}{P(\text{select k})}v_k v_k^\top\right)$$

$$= \mathbf{E}_{B_m}\left(\sum_{k\in B_m}\frac{1}{P(k\in B_m)}v_k v_k^\top\right)$$

$$= \sum_\Omega P(B_m)\times\sum_{k\in B_m}\frac{1}{P(k\in B_m)}v_k v_k^\top$$

$$= \sum_{B_m\in\Omega}\sum_{k\in B_m} P(B_m\mid k\in B_m)\cdot v_k v_k^\top$$

where $\Omega$ denote all $C_n^m$ possible candidate set with choosing $m$ nodes from $n$ nodes, $P(B_m\mid k\in B_m)$ denotes the probability of selecting $m-1$ nodes into $B_m$ conditioning on $k\in B_m$. Denote $\Omega_k$ as all possible size $m$ candidate sets with node $k$ always in the set. Then

$$\mathbf{E}\left(\frac{1}{p_k}v_k v_k^\top\right) = \sum_{k=1}^n\left(\sum_{B_{m-1}^k\subset\Omega_k} P\left(B_m\mid k\in B_m\right)\right)\cdot v_k v_k^\top = \sum_{k=1}^n v_k v_k^\top = I$$

$$\frac{\epsilon}{(\sum R_i)p_k}v_k v_k^\top = \frac{\epsilon}{(\sum R_i)p_k^* q_k}v_k v_k^\top = \frac{\epsilon}{R_k q_k}v_k v_k^\top$$

$$\preceq \epsilon(uI - A)\frac{1}{q_k}$$

$$\leq \frac{m\epsilon}{\eta}(uI - A)$$

where we use the fact $vv^T \preceq (v^T B^{-1}v)B$ for any semi-positive definite matrix $B$. In addition,

$$\mathbf{E}\left(\frac{\epsilon}{(\sum R_i)p_k}v_k v_k^\top\right) = \frac{\epsilon}{\sum R_i}I = \frac{\epsilon}{\Phi_{u,l}(A)}I$$

for any $k = 1,\cdots,n$. Denote $w_k = \sqrt{\frac{\epsilon}{\sum R_i p_k}}v_k$, then $w_k w_k^\top \preceq \frac{m\epsilon}{\eta}(uI - A)$, which implies for any $k\in[1,n]$

$$w_k^\top(uI - A)^{-1}w_k w_k^\top(uI - A)^{-1}w_k \leq \frac{m\epsilon}{\eta}w_k^\top(uI - A)^{-1}w_k$$

$$\Rightarrow\quad w_k^\top(uI - A)^{-1}w_k \leq \frac{m\epsilon}{\eta}$$

Similarly, we have

$$w_k^\top(A - lI)^{-1}w_k \leq \frac{m\epsilon}{\eta}$$

Then from Lemma 3.3 and Lemma 3.4 in [? ] we have

$$\text{Tr}(uI - A - w_k w_k^\top) \leq \text{Tr}(uI - A) + \frac{w_k^\top(uI - A)^{-2}w_k}{1 - \frac{m\epsilon}{\eta}} \tag{19}$$

$$\text{Tr}(A + w_k w_k^\top - lI) \leq \text{Tr}(A - lI) - \frac{w_k^\top(A - lI)^{-2}w_k}{1 + \frac{m\epsilon}{\eta}} \tag{20}$$

$$\tag{21}$$

Define $\epsilon' = \frac{m\epsilon}{\eta}$, we show in the following that $\mathbf{E}\left(\Phi_{u_j,l_j}(A_j)\right) \leq \Phi_{u_{j-1},l_{j-1}}(A_{j-1})$.

From (19) we have

$$\Phi_{u_j,l_j}(A_j) \leq \Phi_{u_j,l_j}(A_{j-1}) + \frac{w_{j-1}^\top(u_j I - A_{j-1})^{-2}w_{j-1}}{1 - \epsilon} - \frac{w_{j-1}^\top(A_{j-1} - l_j I)^{-2}w_{j-1}}{1 + \epsilon}\;\text{\textcircled{3}}$$

Define $\Delta_u = u_j - u_{j-1} = \frac{\epsilon}{(1-\epsilon')\sum R_j}$ and $\Delta_l = l_j - l_{j-1} = \frac{\epsilon}{(1+\epsilon')\sum R_j}$. Notice that

$$\frac{\partial}{\partial u}\mathrm{Tr}(uI - A)^{-1} = -\mathrm{Tr}(uI - A)^{-2} < 0$$

$$\frac{\partial}{\partial l}\mathrm{Tr}(A - lI)^{-1} = \mathrm{Tr}(A - lI)^{-2} < 0$$

at each step based on the design of $u_j$ and $l_j$. and $\Phi_{u,l}(A)$ is convex in terms of $u$ and $l$. From $\textcircled{3}$, we have

$$\Phi_{u_j,l_j}(A_j) \le \Phi_{u_j,l_j}(A_{j-1}) + \frac{1}{1-\epsilon'}\mathrm{Tr}\big[(u_jI - A_{j-1})^{-2}w_{j-1}w_{j-1}^\top\big]$$
$$- \frac{1}{1+\epsilon'}\mathrm{Tr}\big[(A_{j-1} - l_jI)^{-2}w_{j-1}w_{j-1}^\top\big]$$

then with $\mathbf{E}(w_k w_k^T) = \frac{\epsilon}{\sum R_i}I$

$$\mathbf{E}\left(\Phi_{u_j,l_j}(A_j)\right) \le \Phi_{u_j,l_j}(A_{j-1}) + \frac{\epsilon}{(1-\epsilon')\sum R_i}\mathrm{Tr}\big[(u_jI - A_{j-1})^{-2}\big]$$
$$- \frac{\epsilon}{(1+\epsilon')\sum R_i}\mathrm{Tr}\big[(A_{j-1} - l_jI)^{-2}\big]$$
$$\le \Phi_{u_j,l_j}(A_{j-1}) + \Delta_u\mathrm{Tr}\big[(u_jI - A_{j-1})^{-2}\big]$$
$$- \Delta_l\mathrm{Tr}\big[(A_{j-1} - l_j)^{-2}\big]$$

Define

$$f(t) = \mathrm{Tr}\big[(u_{j-1} + t\cdot\Delta_u)I - A_{j-1}\big]^{-1} + \mathrm{Tr}\big[A_{j-1} - (l_{j-1} + \Delta_l\cdot t)I\big]^{-1}$$

then

$$\frac{\partial f(t)}{\partial t} = -\Delta_u\mathrm{Tr}\big[(u_{j-1} + t\cdot\Delta_u)I - A_{j-1}\big]^{-2} + \Delta_l\mathrm{Tr}\big[A_{j-1} - (l_{j-1} + \Delta_l\cdot t)I\big]^{-2}$$

Since $f(t)$ is convex, we have

$$\left.\frac{\partial f(t)}{\partial t}\right|_{t=1} \ge f(1) - f(0) = \Phi_{u_j,l_j}(A_{j-1}) - \Phi_{u_{j-1},l_{j-1}}(A_{j-1}) \tag{22}$$

Then plugin (22), we have

$$\mathbf{E}\left(\Phi_{u_j,l_j}(A_j)\right) \le \Phi_{u_{j-1},l_{j-1}}(A_{j-1})$$

Notice that for selection

$$\frac{\Delta_{u_j} - \Delta_{l_j}}{\Delta_{u_j}} = \frac{\frac{\varepsilon}{t(1-\varepsilon')} - \frac{\varepsilon}{t(1+\varepsilon')}}{\frac{\varepsilon}{t(1-\varepsilon')}} = \frac{\frac{1}{(1-\varepsilon')} - \frac{1}{(1+\varepsilon')}}{\frac{1}{(1-\varepsilon')}} \le 2\varepsilon'$$

where $t = \sum R_i$. We consider that the selection process stops when at the iteration $k$ that $u_k - l_k \ge 8r_d/\epsilon$. Notice $u_0 = \frac{2r_d}{\varepsilon}, l_0 = \frac{-2r_d}{\varepsilon}$, when stop at $u_k - l_k \ge \frac{8r_d}{\varepsilon}$, we have

$$\frac{u_k - l_k}{u_k} = \frac{(u_0 - l_0) + \sum_{j=0}^{k-1}\left(\Delta_{u_j} - \Delta_{l_j}\right)}{u_0 + \sum_{j=0}^{k-1}\Delta_{u_j}}$$
$$\le \frac{4r_d/\varepsilon + \sum_{j=0}^{k-1}\left(\Delta_{u_j} - \Delta_{l_j}\right)}{2r_d/\varepsilon + (2\varepsilon')^{-1}\sum_{j=0}^{k-1}\left(\Delta_{u_j} - \Delta_{l_j}\right)}$$
$$\le \frac{4r_d/\varepsilon + 4r_d/\varepsilon}{2r_d/\varepsilon + (2\varepsilon')^{-1}4r_d/\varepsilon}$$
$$= \frac{8r_d/\varepsilon}{2r_d\left(1 + \frac{1}{\varepsilon'}\right)/\varepsilon}$$
$$= \frac{4}{1 + \frac{1}{\varepsilon'}} \le 4\varepsilon'$$

Then we have $\frac{u_k}{l_k} = \left(1 - \frac{u_k - l_k}{u_k}\right)^{-1} \leq 1 + 4\left(\varepsilon'\right)$. Notice that $u_k - l_k \geq \frac{8r_d}{\varepsilon} \implies \sum_{j=0}^{k-1}\left(\Delta_{u_j} - \Delta_{l_j}\right) \geq \frac{4r_d}{\varepsilon}$.

Consider at the $j$th selection

$$\Delta_{u_j} - \Delta_{l_j} = \left(\frac{\epsilon}{1-\epsilon'} - \frac{\epsilon}{1+\epsilon'}\right)\frac{1}{\sum R_i}$$

$$= \frac{\tilde{\epsilon}}{\Phi_{u_j,l_j}(A_j)}, \quad \epsilon' = \frac{2\epsilon\epsilon'}{(1-\epsilon')(1+\epsilon')}$$

Then

$$P\left(\text{finish selection after } \mathcal{B} \text{ times selection} z \text{ vectors}\right) \geq P\left(\sum_{j=0}^{\mathcal{B}-1}\frac{\tilde{\epsilon}}{\Phi_{u_j,l_j}(A_j)} \geq \frac{4r_d}{\epsilon}\right)$$

$$= P\left(\sum_{j=0}^{\mathcal{B}-1}\Phi_{u_j,l_j}^{-1}(A_j) \geq \frac{4r_d}{\tilde{\epsilon}\epsilon}\right)$$

$$\geq P\left(\frac{\mathcal{B}^2}{\sum_{j=0}^{\mathcal{B}-1}\Phi_{u_j,l_j}(A_j)} \geq \frac{4r_d}{\tilde{\epsilon}\epsilon}\right)$$

$$= P\left(\sum_{j=0}^{\mathcal{B}}\Phi_{u_j,l_j}(A_j) \leq \frac{\mathcal{B}^2\tilde{\epsilon}\epsilon}{4r_d}\right),$$

$$\geq 1 - \frac{4n}{\mathcal{B}\tilde{\epsilon}},$$

$$\geq 1 - \frac{2r_d}{\mathcal{B}\cdot\frac{m}{\eta}\cdot\epsilon^2}$$

where we use the result that $\mathbf{E}\left(\Phi_{u_j,l_j}(A_j)\right) \leq \Phi_{u_0,l_0}(A_0)$ by recursively using $\mathbf{E}\left(\Phi_{u_j,l_j}(A_j)\right) \leq \Phi_{u_{j-1},l_{j-1}}(A_{j-1})$ and the fact that $\Phi_{u_0,l_0}(A_0) = \epsilon$.

We consider the following reparametrization:

$$\epsilon = \frac{\sqrt{\delta}}{C_0}, \quad 0 < \delta, \quad \text{mid} = 2r_d(1 - m^2\epsilon^2)/(m\epsilon^2).$$

For the $j$th selection, $\alpha_j = \frac{\epsilon}{\Phi_j}\frac{1}{\text{mid}}$. From previous result, we have $\frac{u_k}{l_k} = \left(1 - \frac{u_k - l_k}{u_k}\right)^{-1} \leq 1 + 4\left(\varepsilon'\right)$ with probability $1 - 2/C$ with $\delta = \frac{CC_0^2\eta r_d}{m\mathcal{B}}$. Notice that $u_k = u_0 + \sum_{j=1}^{k}\frac{\epsilon}{(1-\epsilon')\Phi_j}$ and $l_k = l_0 + \sum_{j=1}^{k}\frac{\epsilon}{(1+\epsilon')\Phi_j}$, and

$$u_k + l_k = \sum_{j=1}^{k}\frac{\epsilon}{\Phi_j}\left(\frac{1}{1-\epsilon'} + \frac{1}{1+\epsilon'}\right)$$

Then if stop at $k$th selection

$$\Phi_k \geq \frac{2r_d}{u_k - l_k} = \frac{2r_d}{(u_{k-1} - l_{k-1}) + \frac{\epsilon}{\Phi_k}\left(\frac{1}{1-\epsilon'} - \frac{1}{1+\epsilon'}\right)}$$

$$\geq \frac{2r_d}{\frac{8r_d}{\epsilon} + \frac{\epsilon}{\Phi_k}\left(\frac{1}{1-\epsilon'} - \frac{1}{1+\epsilon'}\right)}$$

Denote $c = \frac{1}{1-\epsilon'} - \frac{1}{1+\epsilon'}$, then $\Phi_k \geq \frac{1}{4}\epsilon - \frac{c}{8}\epsilon^2$. We find $C_0$ such that $c\epsilon = \frac{2\epsilon'\epsilon}{1-(\epsilon')^2} < 1$ then $\Phi_k \geq \frac{\epsilon}{8}$

Therefore,

$$u_k - l_k = u_{k-1} - l_{k-1} + \frac{\epsilon}{\Phi_k}\left(\frac{1}{1-\epsilon'} - \frac{1}{1+\epsilon'}\right) \leq \frac{8r_d}{\epsilon} + \frac{\epsilon}{\Phi_k}\left(\frac{1}{1-\epsilon'} - \frac{1}{1+\epsilon'}\right)$$

$$\leq \frac{8r_d}{\epsilon} + 8\left(\frac{1}{1-\epsilon'} - \frac{1}{1+\epsilon'}\right)$$

$$= \frac{8r_d}{\epsilon} + \frac{16\epsilon'}{1-(\epsilon')^2}$$

Then we choose $C_0$ large such that $\frac{16\epsilon'}{1-(\epsilon')^2} < \frac{r_d}{\epsilon} \Rightarrow u_k - l_k \leq \frac{9r_d}{\epsilon}$. Given that $\epsilon' = \frac{m}{\eta}\epsilon$ and $\epsilon = \frac{\sqrt{\delta}}{C_0}$, we choose appropriate $C_0$ to satisfy previous requirement on $\epsilon$ and $\epsilon'$ as

$$\begin{cases} 2\epsilon\epsilon' < 1 - (\epsilon')^2 \\ \dfrac{16\epsilon'}{1-(\epsilon')^2} < \dfrac{r_d}{\epsilon} \\ \epsilon' < 1 \end{cases}$$

Therefore, we choose $C_0$ such that $C_0^2 > \left(\frac{m}{\eta}\right)^2 + \max(2, \frac{16}{r_d}) \times \frac{m}{\eta}$. Notice that

$$u_k - l_k = \sum_{j=1}^{k} \frac{\epsilon}{\Phi_j}\left(\frac{1}{1-\epsilon'} - \frac{1}{1+\epsilon'}\right) \geq \frac{4r_d}{\epsilon}$$

and mid is defined as mid $= \frac{\frac{4d}{\epsilon}}{\frac{1}{1-\epsilon'} - \frac{1}{1+\epsilon'}}$, then $\sum_{j=1}^{k} \frac{\epsilon}{\Phi_j} \geq$ mid. Also,

$$\sum_{j=1}^{k-1} \frac{\epsilon}{\Phi_j} \leq \text{mid} \Rightarrow \text{mid} > \sum_{j=1}^{k} \frac{\epsilon}{\Phi_j} - 8$$

$$> \frac{\epsilon}{\Phi_j} - \frac{4\epsilon\epsilon'}{r_d(1-(\epsilon')^2)}\sum_{j=1}^{k} \frac{\epsilon}{\Phi_j}$$

$$= \left(1 - \frac{4\epsilon\epsilon'}{r_d(1-(\epsilon')^2)}\right)\sum_{j=1}^{k} \frac{\epsilon}{\Phi_j}$$

which implies

$$\text{mid} \in \left[1 - \frac{4\epsilon\epsilon'}{r_d(1-(\epsilon')^2)}, 1\right] \cdot \sum_{j=1}^{m} \frac{\epsilon}{\Phi_j} = \left[1 - \frac{4\epsilon\epsilon'}{r_d(1-(\epsilon')^2)}, 1\right] \cdot \frac{u_k + l_k}{\frac{1}{1-\epsilon'} + \frac{1}{1+\epsilon'}}$$

Notice that for the design matrix $A$ in (18), we have $\frac{1}{\sqrt{\text{mid}}}A = A_k$ where $A_k$ is the accumulated covariance matrix when the query process stops at the $k$the selection. Therefore, the eigenvalues of $A$ satisfy $\lambda(A^T A) = \frac{1}{\text{mid}}\lambda(A_k^T A_k) \in \left[\frac{l_k}{\text{mid}}, \frac{u_k}{\text{mid}}\right]$. Then

$$\left[\frac{l_k}{\text{mid}}, \frac{u_k}{\text{mid}}\right] \subset \left[\frac{l_k}{\frac{u_k+l_k}{\frac{1}{1-\epsilon'}+\frac{1}{1+\epsilon'}}}, \frac{u_k}{\left(1 - \frac{4\epsilon\epsilon'}{r_d(1-(\epsilon')^2)}\right) \cdot \frac{u_k+l_k}{\frac{1}{1-\epsilon'}+\frac{1}{1+\epsilon'}}}\right]$$

Given that with high probability, $1 - 4\epsilon' \leq \frac{u_k}{l_k} \leq 1 + 4\epsilon'$. Then for the lower bound,

$$\frac{l_k}{\frac{u_k+l_k}{\frac{1}{1-\epsilon'}+\frac{1}{1+\epsilon'}}} \geq \frac{1}{(1-(\epsilon')^2)(1+2\epsilon')} > \frac{1}{(1+\epsilon')(1+2\epsilon')} > \frac{1}{2} \times \frac{1}{(1+\epsilon')^2} = \frac{1}{2}\frac{1}{(1+m\sqrt{\delta}/(\eta C_0))^2}$$

and upper bound

$$\frac{u_k}{\left(1 - \frac{4\epsilon\epsilon'}{r_d(1-(\epsilon')^2)}\right) \cdot \frac{u_k+l_k}{\frac{1}{1-\epsilon'} + \frac{1}{1+\epsilon'}}} \leq \frac{1+4\epsilon'}{(1+2\epsilon')(1-(\epsilon')^2) - (1+2\epsilon')\frac{4\epsilon\epsilon'}{r_d}}$$

$$\leq \frac{4}{3} \times \frac{1+4\epsilon'}{(1+2\epsilon)(1-(\epsilon')^2)}, \text{ given } \frac{4\epsilon\epsilon'}{r_d} > \frac{1}{4}(1-(\epsilon')^2)$$

$$< \frac{8}{3} \times \frac{1}{1-(\epsilon')^2}$$

$$< \frac{8}{3} \frac{1}{1-(m\sqrt{\delta}/(\eta C_0))^2}.$$

Then with probability larger than $1 - \frac{2}{C}$, we have

$$\lambda(A^\top A) \in \left[\frac{1}{2} \times \frac{1}{(1 + \frac{m\sqrt{\delta}}{\eta C_0})^2}, \frac{8}{3} \times \frac{1}{1 - (\frac{m\sqrt{\delta}}{\eta C_0})^2}\right]$$

Consider $\alpha_j = \frac{\epsilon}{\Phi_j} \cdot \frac{1}{\text{mid}}$,

$$\Rightarrow \sum_{j=1}^{k} \alpha_j = \sum_{j=1}^{k} \frac{\epsilon}{\Phi_j} \cdot \frac{1}{\text{mid}} \in \left[1, \frac{1}{1 - \frac{4\epsilon\epsilon'}{r_d(1-(\epsilon')^2)}}\right] \leq \frac{1}{1 - \frac{1}{4}\frac{(1-\epsilon')^2}{(1-\epsilon')^2}} = \frac{4}{3}$$

Finally, check $\sup_{j \in \mathcal{S}} \left(s_j \sum_{i=1}^{r_d} |\mathbf{V}_i(j)|^2\right)$ at the $k$th selection

$$\sup_{j \in \mathcal{S}} \left(s_j \sum_{i=1}^{r_d} |\mathbf{V}_i(j)|^2\right) = \sup_{j \in \mathcal{S}} \left\{\frac{\epsilon}{\Phi_k} \cdot \frac{1}{\text{mid}} \times \frac{\Phi_k}{R_j} \times \sum_{i=1}^{r_d} |\mathbf{V}_i(j)|^2\right\}$$

$$= \frac{\epsilon}{\text{mid}} \cdot \sup_j \left\{\frac{\sum_{j=1}^{r_d} |\mathbf{V}_i(j)|^2}{R_j}\right\}$$

$$\leq \frac{\epsilon}{\text{mid}} \cdot \frac{1}{\frac{1}{u_k - l_k} + \frac{1}{u_k - l_k}}$$

$$= \frac{\epsilon}{\text{mid}} \times \frac{u_k - l_k}{2} \leq \frac{\epsilon}{\text{mid}} \times \frac{9 \cdot \frac{r_d}{\epsilon}}{2} = \frac{4.5 r_d}{\text{mid}}$$

$$\leq \frac{4.5 r_d}{\frac{2r_d(1-(\epsilon')^2)}{\epsilon\epsilon'}} = 2.25 \times \frac{\epsilon\epsilon'}{1-(\epsilon')^2} \leq 2.25 \times 4\delta = 10\delta$$

Then we finish the proof of Lemma 1.

## B More on Biased Sequential Sampling

### B.1 Computational complexity

In the representative sampling stage, the computational complexity of calculating the sampling probability is $\mathcal{O}(n)$. We then sample $m$ nodes to formulate a candidate set $B_m$, where the complexity of sampling $m$ variables from a discrete probability distribution is $\mathcal{O}(m)$ [31]. Consequently, the complexity of the representative learning stage is $\mathcal{O}(n + m)$.

In the informative selection stage, we calculate the information gain $\Delta_i$ for each node. This involves obtaining the eigenvector corresponding to the smallest non-zero eigenvalue of the projected graph Laplacian matrix, with a complexity of $\mathcal{O}(n^3)$ due to the singular value decomposition (SVD) operation. Subsequently, we compute $\Delta_i$ for each node in the candidate set $B_m$ based on their loadings on the eigenvector, which incurs an additional computational cost of $\mathcal{O}(mn)$. Therefore, the total complexity of our biased sampling method is $\mathcal{O}(n + m + nm + n^3)$. Given a node label query budget $\mathcal{B}$, the overall computational cost becomes $\mathcal{O}(\mathcal{B}\left(n + m + nm + n^3\right))$.

When the dimension of node covariates $p \ll n$, we can replace the SVD operation with the Lanczos algorithm to accelerate the informative selection stage. The Lanczos algorithm is designed to

efficiently obtain the $k$th largest or smallest eigenvalues and their corresponding eigenvectors using a generalized power iteration method, which has a time complexity of $\mathcal{O}(kn^2)$ [14]. As a result, the complexity of the proposed biased sampling method reduces to $\mathcal{O}(pn^2)$. This is comparable to GNN-based active learning methods, as GNNs and their variations generally have a complexity of $\mathcal{O}(pn^2)$ per training update [6, 37].

## B.2  Connection to classification tasks

Although our theoretical analysis is developed for node regression tasks, the proposed query strategy and graph signal recovery procedure are also applicable to classification tasks. Consider a $K$-class classification problem, where the response on each node $i$ is given by $\mathbf{f}(i) \in 1, 2, \ldots, K$. We introduce a dummy membership vector $(Y_1(i), \ldots, Y_K(i))$, where $Y_c(i) = 1$ if $\mathbf{f}(i) = c$ and $Y_c(i) = 0$ otherwise. For each class $c \in \{1, 2, \ldots, K\}$, we first estimate $\hat{\beta}_c$ based on (14) with the training data $\{\tilde{\mathbf{X}}_{i\cdot}, Y_c(i), s_i\}_{i \in \mathcal{S}}$, and then compute the score for class $c$ as $\hat{\mathbf{f}}_c = \tilde{X}\hat{\beta}_c$. The label of an unqueried node $j$ is assigned as $\hat{\mathbf{f}}(j) = \arg\max_{1 \le c \le K}\{\hat{\mathbf{f}}_1(j), \hat{\mathbf{f}}_2(j), \cdots, \hat{\mathbf{f}}_K(j)\}$. Notice that the above score-based classifier is equivalent to the softmax classifier:

$$\hat{\mathbf{f}} = \arg\max_{1 \le c \le K}\{\frac{\exp(\hat{\mathbf{f}}_1)}{\sum_c \exp(\hat{\mathbf{f}}_c)}, \cdots, \frac{\exp(\hat{\mathbf{f}}_K)}{\sum_c \exp(\hat{\mathbf{f}}_c)}\}$$

since the softmax function is monotonically increasing with respect to each score function $\{\hat{\mathbf{f}}_c\}_{c=1}^K$.

## B.3  Discussion on Theorem 4.1

Theorem 4.1 is derived using first-order matrix perturbation theory [2] on the Laplacian matrix $\mathcal{L}$. In Theorem 4.1, we assume that the column space of the node covariate matrix $\mathbf{X}$ is identical to the space spanned by the first $d$ eigenvectors of $\mathcal{L}_{\mathcal{S}^c}$. This assumption simplifies the analysis and the results by focusing on the perturbation of $\mathcal{L}_{\mathcal{S}^c}$, where $\mathcal{L}_{\mathcal{S}^c}$ is the reduced Laplacian matrix with zero entries in the rows and columns indexed by $\mathcal{S}$.

The analysis can be naturally extended to the general setting by replacing $\mathcal{L}_{\mathcal{S}^c}$ with $\mathbf{P}(\mathbf{1}_{\mathcal{S}^c})\mathcal{L}\mathbf{P}(\mathbf{1}_{\mathcal{S}^c})$, where $\mathbf{P}(\mathbf{t})$ is the projection operator defined in Section 3.2. Moreover, under the assumption on the node covariates, the information gain $\Delta_i$ exhibits an explicit dependence on the network statistics, providing a clearer interpretation of how the network structure influences the benefits of selecting informative nodes.

Theorem 4.1 indicates that the improvement of biased sampling is more significant when $d_{\min}$ is larger and $\eta_0, \eta_1$ are smaller. Specifically, $d_{\min}$ reflects the connectedness of the network, where a better-connected network facilitates the propagation of label information and enhances the informativeness of a node's label for other nodes. A smaller $\eta_1$ prevents the existence of dominating nodes, ensuring that the connectedness does not significantly decrease when some nodes are removed from the network.

Notice that the node $j^*$ is the most informative node for the next selection, and $\eta_0$ measures the number of nodes similar to $j^*$ in the network. Recall that the proposed biased sampling method considers both the informativeness and representativeness of the selected nodes. Therefore, the information gain is less penalized by the representativeness requirement if $\eta_0$ is small. Additionally, the size of the candidate set $m$ should be sufficiently large to ensure that informative nodes are included in $B_m$.

## B.4  Discussion on Theorem 4.2

The RHS of (15) captures both the variance and bias involved in estimating $\mathbf{f}$ using noisy labels on sampled nodes. Specifically, the first three terms represent the estimation variance arising from controlling the condition number of the design matrix on the queried nodes. The fourth and fifth terms reflect the noise and unidentifiable components in the responses of the queried nodes, while the last term denotes the bias resulting from the approximation error of the space using $\mathbf{H}_\omega(\mathbf{X}, \mathbf{A})$.

The bias term in Theorem 4.2 can be further controlled if the true signal $\mathbf{f}$ exhibits decaying or zero weights on high-frequency network components. In addition to $r_d$, the size of the candidate set $m$ also influences the probability of controlling the generalization error. A small $m$ places greater emphasis

on the representativeness criterion in sampling, increasing the likelihood of controlling the condition number but potentially overlooking informative nodes, thereby increasing approximation bias.

For a fixed prediction MSE, the query complexity of our method is $\mathcal{O}(d)$, whereas random sampling incurs a complexity of $\mathcal{O}(\tilde{d} \log \tilde{d})$, where $\tilde{d} > d$. Our method outperforms random sampling in two key aspects: (1) The information-based selection identifies $\mathbf{f}$ with fewer queries than random sampling, as shown in Theorem 4.1, and (2) our method achieves an additional improvement by actively controlling the condition number of the covariate matrix, resulting in a logarithmic factor reduction compared to random sampling.

### B.5 Calculation on node-wise information gain and hyperparameter tuning

When calculating node-wise informativeness in (7), we can enhance computational efficiency by avoiding the inversion of $D(\mathbf{t})$. When $\mathbf{t}$ is in the neighborhood of $\mathbf{1}_{\mathcal{S}^c}$, we can approximate:

$$\mathbf{P}(\mathbf{t}) \approx D(\mathbf{t})\mathbf{X}_{\mathcal{S}^c}(\mathbf{X}_{\mathcal{S}^c}^T \mathbf{X}_{\mathcal{S}^c})^{-1}\mathbf{X}_{\mathcal{S}^c}^T D(\mathbf{t}) = D(\mathbf{t})\mathbf{Z}_{\mathcal{S}^c}\mathbf{Z}_{\mathcal{S}^c}^T D(\mathbf{t}),$$

where $\mathbf{X}_{\mathcal{S}^c} = ((X_1)_{\mathcal{S}^c}, \cdots, (X_p)_{\mathcal{S}^c})$ and $\mathbf{Z}_{\mathcal{S}^c} = \mathbf{X}_{\mathcal{S}^c}(\mathbf{X}_{\mathcal{S}^c}^T \mathbf{X}_{\mathcal{S}^c})^{-1/2}$. Then, the node-wise informativeness can be explicitly expressed as:

$$\Delta_i \propto t_i \phi_i^2 (\mathbf{Z}_{\mathcal{S}^c})_i. (\mathbf{Z}_{\mathcal{S}^c}^T)_i. + \sum_{j \neq i, 1 \leq j \leq n} t_i t_j \phi_i \phi_j (\mathbf{Z}_{\mathcal{S}^c})_i. (\mathbf{Z}_{\mathcal{S}^c}^T)_j. \tag{23}$$

We find that this approximation yields very similar empirical performance compared to the exact formulation in (7). Therefore, we adopt the formulation in (23) for the subsequent numerical experiments.

In practice, we can tune $m$ to ensure that the covariance matrix is well-conditioned. Specifically, we can run the biased sampling procedure multiple times with different values of $m$ and select the largest $m$ such that the condition number of the covariance matrix on the query set $\mathcal{S}$ is less than 10 [20]. This threshold is a commonly accepted rule of thumb for considering a covariance matrix to be well-conditioned [20]. Additionally, $\epsilon$ is typically fixed at a small value, following the protocol outlined in [17].

## C  More on Numerical Studies

### C.1  Experimental setups

**Synthetic networks**  The parameters for the three network topologies are: Watts–Strogatz (WS) model ($K = 4, \beta_{WS} = 0.1$) for small world properties, Stochastic block model (SBM) ($N_{\text{community}} = 4, P_{\text{in}} = 0.35, P_{\text{out}} = 0.01$) for community structure, and Barabási-Albert (BA) model ($\alpha = 3$) for scale-free properties. We set $n = 100$ for all three networks. After generating the networks, we consider them fixed and then simulate $\mathbf{Y}$ and $\mathbf{X}$ repeatedly using 10 different random seeds. By a slight abuse of notation, we set the node responses and covariates for SBM and WS as $\mathbf{Y} = U_{1:10}\beta + \xi$ and $\mathbf{X} = U_{1:10} + MU_{45:54}$, where $M_{ij} \overset{\text{iid}}{\sim} N(0.3, 0.1)$ and $\beta = (\underbrace{5, 5, \ldots, 5}_{\text{length 10}})^T$. For the BA model, we set $\mathbf{Y} = U_{1:15}\beta + \xi$ and $\mathbf{X} = U_{1:15} + MU_{45:59}$, where $M_{ij} \overset{\text{iid}}{\sim} N(0.5, 0.2)$ and $\beta = (\underbrace{1, \ldots, 1}_{\text{length 5}}, \underbrace{5, \ldots, 5}_{\text{length 10}})^T$.

**Real-world networks**  For the proposed method and all baselines, we train a 2-layer SGC model for a fixed 300 epochs. In SGC, the propagation matrix performs low-pass filtering on homophilic networks and high-pass filtering on heterophilic networks. During training, the initial learning rate is set to $10^{-2}$ and weight decay as $10^{-4}$.

### C.2  Visualization

In Figure 2, we visualize the node query process on synthetic networks generated using SBM and BA, as described in Section 5.1. The figure clearly demonstrates that nodes queried by the proposed algorithm adapt to the *informativeness* criterion specific to each network topology, effectively aligning with the community structure in SBM and the scale-free structure in BA.

| Dataset | #Nodes | Type | $m$ | $d$ | Dataset | #Nodes | Type | $m$ | $d$ |
|---------|--------|------|-----|-----|---------|--------|------|-----|-----|
| SBM | 100 | homophilic | 50 | 10 | Citeseer | 3,327 | homophilic | 1000 | 100 |
| WS | 100 | homophilic | 50 | 10 | Chameleon | 2,277 | heterophilic | 800 | 30, 30 |
| BA | 100 | homophilic | 50 | 15 | Texas | 183 | heterophilic | 60 | 15, 15 |
| Cora | 2,708 | homophilic | 2000 | 200 | Ogbn-Arxiv | 169,343 | homophilic | 1000 | 120 |
| Pubmed | 19,717 | homophilic | 3000 | 60 | Co-Physics | 34,493 | homophilic | 3000 | 150 |

Table 4: A description of all datasets used in Section 5 and the hyperparameter settings for each dataset. We set $\epsilon = 0.001$ for all networks. For heterophilic networks, we combine the eigenvectors corresponding to the $d$ smallest and $d$ largest eigenvalues.

## C.3  Ablation study

To gain deeper insights into the respective roles of representative sampling and informative selection in the proposed algorithm, we conduct additional experiments on a New Jersey public school social network dataset, **School** [25], which was originally collected to study the impact of educational workshops on reducing conflicts in schools. As **School** is not a benchmark dataset in the active learning literature, we did not compare the performance of our method against other baselines in Section 5.2 to ensure fairness. In this dataset with $n = 615$ nodes, each node represents an individual student, and edges denote friendships among students. We treat the students' grade point averages (GPA) as the node responses and select $p = 5$ student features—grade level, race, and three binary survey responses—as node covariates using a standard forward selection approach.

As shown in Section 3.4, the representative sampling in steps 1 and 2 of Algorithm 1 is essential to control the condition number of the design matrix and, consequently, the prediction error given noisy network data. We illustrate in Figure 3a the condition number $\frac{\lambda_{max}(\tilde{X}_{\mathcal{S}}^T W_S \tilde{X}_{\mathcal{S}})}{\lambda_{min}(\tilde{X}_{\mathcal{S}}^T W_S \tilde{X}_{\mathcal{S}})}$ using the proposed method, and compare with the one using random selection. With $m = 200$, the proposed algorithm achieve a significantly lower condition number than random selection, especially when the number of query is small. In the Citeseer dataset, we investigate the prediction performance of Algorithm 1 when removing steps 1 and 2, i.e., setting the candidate set $B_m = \mathcal{S}_{t-1}^c$ for the $t^{th}$ selection. Figure 3b shows that, with representative sampling, the Macro-F1 score is consistently higher, with a performance gap of up to 15%. Given that node classification on Citeseer is found to be sensitive to labeling noise [38], this result validates the effectiveness of representative sampling in improving the robustness of our query strategy to data noise.

In addition, we examine the ability of the proposed method to integrate node covariates for improving prediction performance. In the School dataset, we compare our method to one that removes node covariates during the query stage by setting $\mathbf{X}$ as the identity matrix $\mathbf{I}$. Figure 3c illustrates that the prediction MSE for GPA is significantly lower when incorporating node covariates, thus distinguishing our node query strategy from existing graph signal recovery methods [12] that do not account for node covariate information.

## C.4  Code

The implementation code for the proposed algorithm is available at github.com/Yuanchen-Wu/RobustActiveLearning/.

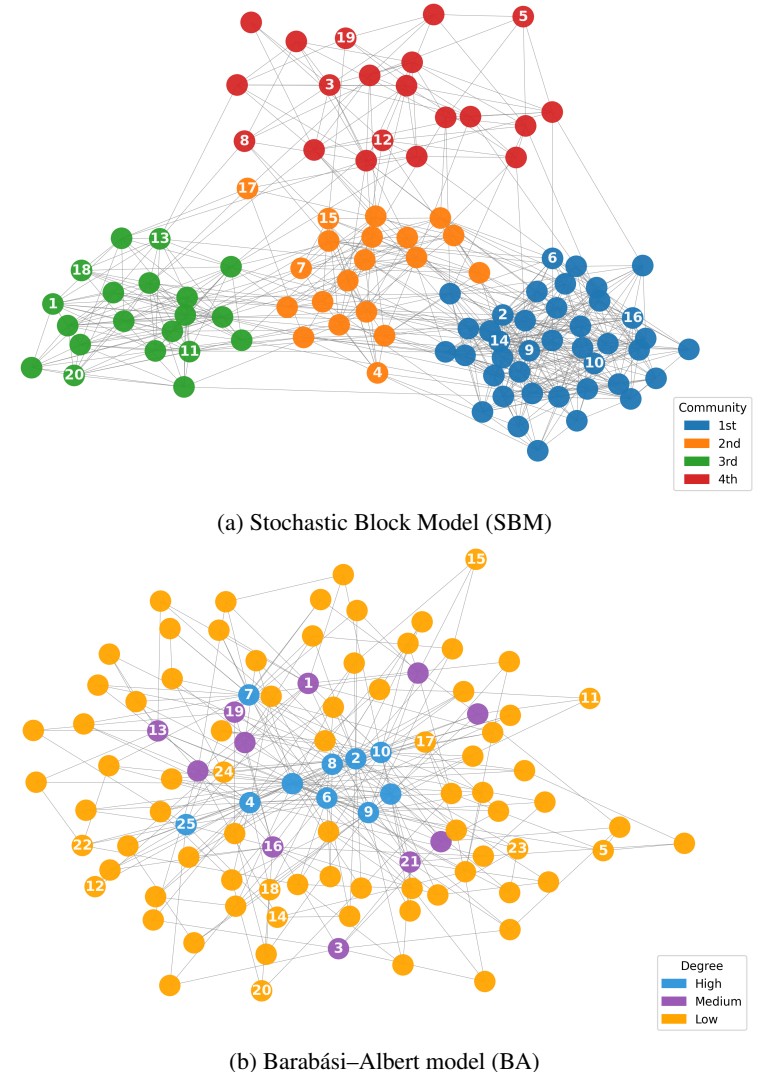

(a) Stochastic Block Model (SBM)

(b) Barabási–Albert model (BA)

Figure 2: For (a) SBM, nodes are grouped by the assigned community; for (b) BA, nodes are grouped by degree. The integer $i$ on each node represents the $i^{th}$ node queried by the proposed algorithm in one replication.

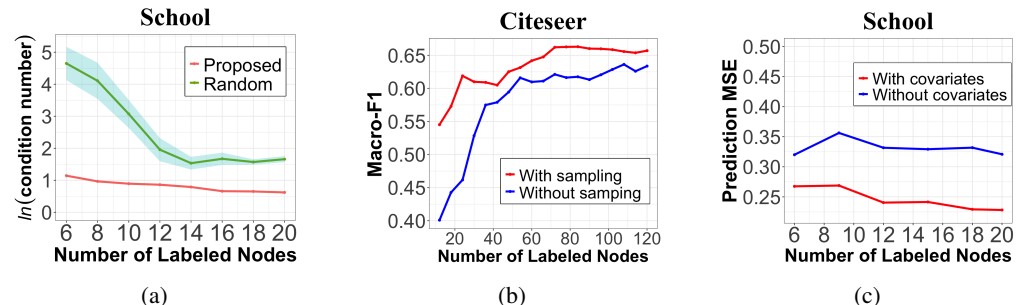

Figure 3: Ablation study: (a) The condition number (log scale) of the design matrix of query nodes selected by proposed method and random sampling. The effectiveness of (b) representative sampling and (c) incorporating covariate information in Algorithm 1.

