# OpenReview forum: "Robust Offline Active Learning on Graphs"
_NeurIPS.cc/2024/Conference — NeurIPS 2024 poster_

### Official Review · Reviewer_hH2h · 2024-07-12

**Soundness:** 2
**Presentation:** 2
**Contribution:** 2
**Rating:** 5
**Confidence:** 3

**Summary:**

This paper proposes an offline active learning method that selects nodes to query by explicitly incorporating information from both the network structure and node covariates. This paper establishs a theoretical relationship between generalization error and the number of nodes selected by the proposed method.

**Strengths:**

1. The theoretical analysis is sufficient.

2. Offline graph active learning is important.

3. The proposed method is easy to implement.

**Weaknesses:**

I appreciate the author's theoretical analysis, but the experimental section is clearly insufficient. Recent work on graph active learning[1] has been conducted on larger-scale datasets like Arxiv and Products. Conducting experiments solely on datasets such as Cora is not adequate.

[1] Partition-based active learning for graph neural networks. TMLR 2023.

**Questions:**

Please provide detailed experimental results for additional datasets.

**Limitations:**

The authors adequately addressed the limitations and potential negative societal impact of their work.

---

> ### Author Rebuttal · Authors · 2024-08-06
>
> Thank you for acknowledging the theoretical analysis of our algorithm and for your insightful suggestions! We have carefully considered your concerns regarding the insufficient experimental results, and have worked diligently to address them.
>
> >**Weakness: Recent work on graph active learning[1] has been conducted on larger-scale datasets like Arxiv and Products. Conducting experiments solely on datasets such as Cora is not adequate.**
>
> To demonstrate the scalability of our algorithms on larger-scale datasets, we conducted experiments on the two largest datasets (Co-Physics with n=34,493 and Ogbn-Arxiv with n=169,343) included in your suggested paper [1]. The results of our algorithm and the baseline methods are summarized in **Table 2** of the global response PDF.  The greatest improvement is observed in the Macro-F1 score on Arxiv, with a margin as large as 4.8% at 320 labeled nodes. We argue that this is particularly significant given that Arxiv (41 classes) is a class-imbalanced data, where Macro-F1 is a more appropriate metric for evaluation.
>
> Inspired by your comment, we carefully examine computational cost of the proposed method. The complexity of our method is $\mathcal{O}(n+m+nm+n^3)$ for a single node query. When dimension of node feature $p<<n$, we can speed up the informative selection via replacing the SVD by Lanczos algorithm to obtain the $p$th largest or smallest eigenvalues and the corresponding eigenvectors. The time complexity of Lanczos algorithm is $\mathcal{O}(pn^2)$ [2]. Then the complexity of the proposed biased sampling method is $\mathcal{O}(pn^2)$ for single node query. This complexity is comparable to GNN-based network active learning methods since GNN in general has complexity $\mathcal{O}(pn^2)$ in single training update [3].
>
> More interestingly, we found that there is no need to store and perform SVD on the $n$-by-$n$ $P_{\mathcal{S}^c} L^k P_{\mathcal{S}^c}$ to obtain its eigenvectors and eigenvalues for node selection.  Notice that for the rank-p projection matrix
> $P_{\mathcal{S}^c} = Z_{\mathcal{S}^c} Z_{\mathcal{S}^c}^T$ where $Z_{\mathcal{S}^c} \in \mathcal{R}^{n\times p}$ is the base of $P_{\mathcal{S}^c}$. Then we can first perform SVD on $p$-by-$p$ matrix $Z_{\mathcal{S}^c}^T L^k  Z_{\mathcal{S}^c} = U^T\Sigma U$, and the desired eigenvalues and eigenvectors are $\Sigma$ and $Z_{\mathcal{S}^c}U^T$, respectively. During the process, we only need to store and SVD a $p$-by-$p$ and a $n$-by-$p$ matrix, which can be handled efficiently via GPU-based matrix multiplication even when $n$ is large. We report the computational time of proposed method for one node query on multiple benchmark network data in **Table 3** of the PDF. The time cost of single querying is about 2 second when $n$ is about 170,000.
>
> To the best of our knowledge, we did not find offline graph-based active learning methods in the current literature have been tested on the Products dataset (n=1,569,960) . We admit that it is difficult to re-run our method and all benchmark methods on this dataset within the limited rebuttal period. However, we appreciate the reviewer for pointing out this interesting dataset, and we will include these results in the final version of our paper.
>
> >**Question: Please provide detailed experimental results for additional datasets.**
>
> In addition to larger-scale datasets, we also included additional datasets that cover a wide range of homophily and heterophily levels. Besides the benchmark homophily networks (Cora, Citeseer, and Pubmed), we conducted experiments on two heterophily networks (Texas and Chameleon). We also ran all the competitive baselines on these two datasets, as they were not included in the original papers of any of the baselines. The results are summarized in **Table 1** of the global response PDF. Our algorithm achieves the best performance in Cora and Texas and is comparable to the best baselines in Citeseer, Pubmed, and Chameleon.
>
> Moreover, we conducted simulation studies on synthetic networks with three different topologies: small-world property, community structure, and scale-free property. The results are summarized in **Figure 4** of the global response PDF. The proposed algorithm achieved the best performance in all three scenarios under different noise levels.
>
> **Summary:** Following your questions, we conducted experiments on additional networks of much greater size, different levels of homophily, and various topologies. The proposed algorithm achieved competitive, if not the best, performance in  every category, indicating its scalability, generalizability, and robustness.
>
> >**References**
>
> [1] Partition-based active learning for graph neural networks. TMLR 2023.\
> [2] Golub, Gene H., and Charles F. Van Loan. Matrix computations. JHU press, 2013.\
> [3] Wu, Zonghan, et al. "A comprehensive survey on graph neural networks." IEEE transactions on neural networks and learning systems 32.1 (2020): 4-24.

---

> > ### Comment · Reviewer_hH2h · 2024-08-10
> >
> > Thank you for the detailed response. I raised my score to 5.

---

> > > ### Author Response · Authors · 2024-08-10
> > > **Thank you for raising the score!**
> > >
> > > We appreciate your recognition of the updated results and the improved score. If there are any other comments or questions, we would be pleased to discuss and clarify further!

---

### Official Review · Reviewer_7FJA · 2024-07-12

**Soundness:** 3
**Presentation:** 3
**Contribution:** 3
**Rating:** 6
**Confidence:** 2

**Summary:**

The paper proposes a strategy for collecting labeled data for a semi-supervised learning algorithm focused specifically on learning on graphs. The paper provides a theoretical analysis of the proposed method, capturing both the quality of the samples that are selected for labeling as well as the the prediction error of the overall learning procedure. Experimental results show that the method the applicability of the method to real-world datasets.

**Strengths:**

- The method proposed in the paper, as well as the problem setting are explained clearly.
- The paper provides theoretical guarantees for the method that indicate its superiority compared to random sampling (Theorem 2) and characterize the error rate that can be achieved with this active SSL strategy (Theorem 3).

**Weaknesses:**

- The empirical analysis does not very convincingly suggest that the proposed method is better than the baselines considered. Moreover, it would be helpful if the figures showed confidence intervals or error bars.
- The experimental results only compare with a few heuristics for data collection. It would be informative to consider other AL works proposed in the graph learning literature as baselines.
- It is not very clear how the paper is positioned in the literature. It would help to have a related work section that can indicate prior works on AL and active SSL on graphs.
- The clarity of the section 3 could potentially be improved, perhaps by reducing the amount of symbols to the ones that are strictly necessary and providing more clearly marked (e.g. with paragraph titles) intuitive descriptions of the steps that need to be taken and the obstacles that need to be overcome.

Minor remarks:
- lines 19-21: 3 different learning paradigms are mentioned in the first two sentences (active, semi-supervised and transductive learning). It would help if it was clearer early on how they are relevant for the problem that motivates this work.
- line 89: undefined symbol $\mathcal{B}$

**Questions:**

- What is the computational efficiency of the proposed method?
- How tight is the upper bound in Theorem 3? Would it be possible to compare it to numbers from simulations on some simple synthetic settings?
- How does the method compare to online AL methods for graphs (e.g. [36, 37, 30] etc) or other offline AL methods for graphs?

**Limitations:**

While the paper discusses some of the limitations of the proposed approach, it would be good to have a more detailed section on the computation cost of running the method as well as show the impact of various hyperparameters of the method on performance (e.g. m, various properties of the network and the generating process for (X, Y) etc).

---

> ### Author Rebuttal · Authors · 2024-08-07
>
> We thank the reviewer for the constructive feedback, which greatly improves our paper! We address the reviewer's comments point by point below.
>
> >**Weakness 1: empirical analysis does not very convincingly**
>
> Thank you for the comment! In the global response PDF, we included additional experiments on networks with various topologies (**Figure 4**) , varying levels of homophily (**Table 1**) and much larger scales (**Table 2**) . The proposed algorithm achieved the best prediction performance on three synthetic networks with different topologies, the large-scale network Arxiv, the homophily network Cora, and the heterophily network Texas.  For the other four datasets, it is fair to argue that our performance was also competitive. Given these promising results, we are excited to refine the current algorithm in future work, such as extending it to the online setting, to further enhance its empirical performance on different networks. In the updated numerical results, we have included **error bars** indicating standard deviation following your suggestion.
>
> >**Weakness 2:  compare with AL baselines**
>
> We have included several more SoTA offline methods (RIM, GPT, SPA , FeatProp) and online methods (AGE, IGP)  graph active learning methods in the additional experiments on both synthetic and real-world networks. Please refer to **Experiments** in the global response for details.
>
> > **Weakness 3:  literature review on related works on graph-based active learning**
>
> Many graph-based active learning strategies have been proposed based on the principle of maximizing query gain across various information criteria that are defined on the graph domain. The effectiveness of maximizing the graph-domain-based information measurements is generally not guaranteed and challenging to analyze due to quantify complexity of graph signal on graph domain. While complexity measure of binary functions have been proposed for the graph domain [1], its extension to general graph signal with node-wise features remains unclear. Without an analyzable complexity measurement of the labeling function, information maximization may not align with the fastest direction of searching labeling function space. Moreover, most of the existing query information measurements rely on real-time label feedback and are not appliable in offline batch settings.
>
> We propose a new active learning method on spectral domain based on graph spectral method. While spectral methods are utilized in graph sampling and signal reconstruction task [2,3], we utilize spectral methods to introduce a well-defined complexity measurement of labeling function and the associated query strategy in the spectral domain.
>
> Our method is also related to the active learning for regression problem, where learning performance is guaranteed via sample complexity analysis [4,5]. The most notable solution is to use importance sampling based on statistical leverage scores [6], which has sub-optimal sample complexity.  The sample complexity of active regression problems is studied under 𝑙𝑝 norm loss function [7]. Recently, it has been shown that the optimal sample complexity can be linear in terms of the number of regression parameters [8]. Existing methods along this line focus on linear regression [7, 8] or polynomial regression [9], whereas our method extends the theoretical guarantee of active regression learning to graph semi-supervised learning task.
>
>
> > **Weakness 4: better presentation for section 3**
>
> We appreciate reviewer's feedback. We have revised the notation system in this paper to simplify and better present the proposed method. In addition, we re-organize the materials and add intuitive discussion to enhance the logic flow and readability.
>
> >**Weakness minor**:
>
> We thank the reviewer for thoughtful comments. We have re-organized materials in introduction, and clearly define $\mathcal{B}$ as the query budget before line 89.
>
> > **Q1: computational efficiency of the proposed method**
>
> Please see global response on **computational cost**.
>
> > **Q2: tightness of the upper bound in Theorem 3 and empirical validation**
>
> With $d$ and $m$ fixed, Theorem 3 implies MSE decays at a rate of $\frac{1}{\sqrt{\mathcal{B}}}$ where $\mathcal{B}$ is the query budget. We demonstrate the relationship between MSE and $\mathcal{B}$ on simulated data in **Figure 1** of the global response PDF. The simulation demonstrates that the order of sample complexity in Theorem 3 matches the empirical results, implying the tightness of the generalization bound in Theorem 3. Please also see **Optimality of sample complexity** in global response.
>
>
> > **Q3: Compare to online and offline AL methods**
>
> Please see Weakness 2 and **Experiments** in global response.
>
> >**References**
>
> [1] Dasarathy et al. (2015) S2: An efficient graph based active learning algorithm with application to nonparametric classification. COLT
>
> [2] Gadde et al. (2014). Active semi-supervised learning using sampling theory for graph signals. ACM SIGKDD
>
> [3] Shuman et al (2013). The emerging field of signal processing on graphs. IEEE
>
> [4] Kiefer & Wolfowitz (1959). Optimum designs in regression problems. The annals of mathematical statistics
>
> [5] Chaudhuri et al. (2015) Convergence rates of active
> learning for maximum likelihood estimation. NeurIPS
>
> [6] Mahoney et al (2011). Randomized algorithms for matrices and data. Foundations & Trends in Machine Learning
>
> [7] Musco et al (2022). Active linear regression for p norms and beyond. IEEE
>
> [8] Chen & Price (2019). Active regression via linear-sample sparsification. COLT
>
> [9] Meyer et al. (2023) Near-linear sample complexity for lp polynomial regression. ACM

---

> > ### Author Response · Authors · 2024-08-12
> > **Follow-up on our response to your feedback**
> >
> > We are very grateful for the time and effort you have devoted to reviewing our work, and we deeply appreciate your insightful, valuable, and encouraging comments.
> >
> > In response to your questions, we conducted additional experiments to compare the proposed method with SoTA offline graph-based active learning methods (RIM, GPT, SPA, FeatProp), and online methods (AGE, IGP). The numerical comparisons are conducted on networks with various topologies and response noise (Figure 4), different levels of homophily (Table 1), and larger scales (Table 2). In these experiments, the performance of the proposed method is either best or close to the best one. In addition, we follow your suggestion to add error bars when presenting the results.
> >
> > We discuss the optimality of the generalization bound in Theorem 3 and illustrate the tightness via simulation in Figure 1. We also provide a detailed discussion of the computational complexity of the proposed method, which is at $\mathcal{O}(pn^2)$ where $n$ and $p$ are the number of nodes and node features, respectively.
> >
> > We sincerely hope our response adequately addresses your concerns and will definitely incorporate these changes in the revised version. We look forward to your feedback with great anticipation.

---

> > > ### Comment · Reviewer_7FJA · 2024-08-13
> > > **Response to rebuttal**
> > >
> > > I would like to thank the author's for their detailed answers and for the additional experiments. After carefully reading the responses I have decided to maintain my score.

---

> > > > ### Author Response · Authors · 2024-08-13
> > > > **Thank you for the feedback!**
> > > >
> > > > We appreciate your valuable feedback, which has greatly improved our paper. If you have any further comments or questions, we would be more than happy to discuss and provide clarification.

---

### Official Review · Reviewer_emFb · 2024-07-13

**Soundness:** 3
**Presentation:** 3
**Contribution:** 3
**Rating:** 5
**Confidence:** 4

**Summary:**

The paper addresses the challenge of active learning on graphs where labeling node responses is costly. The authors propose an offline active learning method that selects nodes by incorporating both network structure and node covariates. The method leverages graph signal recovery theories and random spectral sparsification, employing a two-stage biased sampling strategy to balance informativeness and representativeness.

**Strengths:**

- The paper introduces a novel offline active learning approach that integrates network structure and node covariates.
- The proposed method is validated through extensive experiments on both synthetic and real-world datasets, showcasing its robustness and effectiveness.

**Weaknesses:**

- How does the proposed method perform on networks with varying levels of homophily and heterophily? Can it adapt to different types of network structures?
- Can the method be extended to online active learning scenarios, where nodes are queried in a sequential manner rather than in batches?
- How does the performance of the proposed method compare with state-of-the-art graph neural network-based active learning methods [1] in a broader range of network topologies and noise conditions?
- Is there a significant computational overhead associated with the two-stage biased sampling strategy, and how does it impact the scalability of the method for large-scale networks?

[1] Focus on Informative Graphs! Semi-supervised Active Learning for Graph-level Classification. Pattern Recognition 2024

**Questions:**

See above.

---

> ### Author Rebuttal · Authors · 2024-08-07
>
> Thank you for acknowledging our algorithm's novelty and providing insightful feedback! We address your concerns point by point below.
> ## Weaknesses:
> >**(A) Performance on networks with varying levels of homophily and heterophily**
>
> Thanks for raising an excellent point about the generalizability of our method to networks with different levels of homophily. Our method can be flexible in adjusting the heterophily in node selection. For example, we can construct space $\bf{H}(\bf{X},\bf{A})$ based on eigenvectors of $\mathcal{L}$ corresponding to large eigenvalues. To handle coexistence of homophily and heterophily, we can adjust the space by combining eigenvectors corresponding to either small or large eigenvalues.
>
> Current methods have primarily been tested on networks with homophily (e.g., Cora, Citeseer). To address your question, we conducted additional experiments and tested competitive baselines on two networks with strong heterophily (Texas and Chameleon). The results, summarized in **Table 2** of the PDF, show that our algorithm achieves the best performance on Cora (strongest homophily) and Texas (strongest heterophily), and is also competitive on other datasets.
>
> >**(B)  Extension to online active learning scenarios**
>
> The proposed method can be directly extended into sequential query manner and online setting as long as the entire network among nodes to be queried are accessible before the query process starts, which guarantees that the function space $\bf{H}(\bf{X},\bf{A})$ is fixed therefore the label information can be accumulated for signal recovery. The biased sampling procedure can select one node and query its label. With the updated set of labelled nodes, we can target a subspace of $\bf{H}(\bf{X},\bf{A})$ that better fits current labels. In next query iteration, we select unlabelled node maximizing information gain on the identified subspace. Intuitively, we gradually shrink the search space during the sequential query, with the goal to find a small but informative function space to estimate graph single, which facilitates to reduce generalization error.
> In summary, the proposed method can be extended into sequential learning scenario, and we reasonably conjecture that the performance of proposed method can be further improved under sequential learning scenario.
>
> >**(C)  Performance with SoTA methods [1] in a broader range of network topologies and noise conditions**
>
> Thank you for the suggestion! We would like to clarify that [1] is designed for graph-level classification task, which differs from our method that aims to node-level classification tasks literature. However, [1] is an interesting read, and we will discuss it in our literature review section.
>
> To address your question, we consider three topologies using synthetic networks:  Watts–Strogatz model for **small world** property, Stochastic Block model for **community structure**, and Barabási-Albert model for **scale-free** property. After generating the network, we simulate observed response $\mathbf{Y}=f+\epsilon$,  where $f$ is weighted linear combination of leading eigenvectors and $\epsilon \sim N(0, \sigma^2 I_n)$ with noise level $\sigma^2\in(0.5, 0.6, 0.7, 0.8, 0.9, 1)$. Based on **Figure 4** of the PDF, the proposed method outperforms SoTA offline active learning methods and is robust to noise.
>
> >**(D) Computational overhead and scalability for large-scale networks**
>
> The complexity of our biased sampling method is $\mathcal{O}(n+m+nm+n^3)$. When the budget of node label query is $\mathcal{B}$, the total computational cost is then $\mathcal{O}(\mathcal{B}(n+m+nm+n^3))$.
>
> The main complexity of the proposed sampling method originates from the SVD operation. When dimension of node feature $p<<n$, we can speed up the informative selection via replacing the SVD by Lanczos algorithm to obtain the $p$th largest or smallest eigenvalues and the corresponding eigenvectors. The time complexity of Lanczos algorithm is $\mathcal{O}(pn^2)$ [2]. Then the complexity of the proposed biased sampling method is $\mathcal{O}(pn^2)$ for single node query. This complexity is comparable to GNN-based network active learning methods since GNN in general has complexity $\mathcal{O}(pn^2)$ in single training update [3].
>
> In terms of memory cost, the main cost is to store the $n$-by-$n$ graph laplacian matrix $\mathcal{L}^k$. However, when the network is sparse, $\mathcal{L}^k$ is also sparse given moderate $k$, which can be handled with memory-efficient sparse matrix formats such as Python SciPy package. More importantly, we do not need to store and SVD the $n$-by-$n$ $P_{\mathcal{S}^c} L^k P_{\mathcal{S}^c}$ to obtain its eigenvectors and eigenvalues for node selection.  Notice that for the rank-p projection matrix
> $P_{\mathcal{S}^c} = Z_{\mathcal{S}^c} Z_{\mathcal{S}^c}^T$ where $Z_{\mathcal{S}^c} \in \mathcal{R}^{n\times p}$ is the base of $P_{\mathcal{S}^c}$. Then we can first perform SVD on $p$-by-$p$ matrix $Z_{\mathcal{S}^c}^T L^k  Z_{\mathcal{S}^c} = U^T\Sigma U$, and the desired eigenvalues and eigenvectors are $\Sigma$ and $Z_{\mathcal{S}^c}U^T$, respectively. During the process, we only need to store and SVD a $p$-by-$p$ and a $n$-by-$p$ matrix, which can be handled efficiently via GPU-based matrix multiplication even when $n$ is very large. We report the computational time of proposed method for one node query on multiple benchmark network data in **Table 3** of the PDF. The time cost of single querying is about 2 second when $n$ is about 170,000.
>
> >**References**
>
> [1] Focus on Informative Graphs! Semi-supervised Active Learning for Graph-level Classification. Pattern Recognition 2024\
> [2] Golub, Gene H., and Charles F. Van Loan. Matrix computations. JHU press, 2013.\
> [3] Wu, Zonghan, et al. "A comprehensive survey on graph neural networks." IEEE transactions on neural networks and learning systems 32.1 (2020): 4-24.

---

> > ### Author Response · Authors · 2024-08-12
> > **Follow-up on our response to your feedback**
> >
> > We sincerely appreciate the time and effort you've put into reviewing our work, and we are truly thankful for your insightful, valuable, and encouraging feedback.
> > In response to your constructive suggestions on numerical experiments, we have conducted additional experiments on real-world networks exhibiting different levels of homophily and heterophily (Table 1), as well as on synthetic networks with different topologies, including community structure, scale-free properties, and small-world properties (Figure 4).
> >
> > As a further supplement to Figure 4, we perform the proposed method with SoTA offline active learning methods on Erdős–Rényi random graph since many social and biological networks can be modeled via this model. The prediction MSE from different methods on **Erdős–Rényi graph** (n=100, p=0.25) with varying response noise are illustrated in the following table:
> >
> > | Method | $\sigma^2$=0.5 | $\sigma^2$=0.6 | $\sigma^2$=0.7 | $\sigma^2$=0.8 | $\sigma^2$=0.9 | $\sigma^2$=1.0 |
> > |------------|-------------------|-------------------|-------------------|-------------------|-------------------|-------------------|
> > | D-Optimal  | 0.48 &plusmn;  0.03            | 0.69  &plusmn;  0.04            | 0.93&plusmn;0.06                 | 1.22 &plusmn; 0.08               |  1.54&plusmn;      0.1       | 1.9 &plusmn; 0.12         |
> > | SPA        | 0.61 &plusmn;  0.05          | 0.88 &plusmn;  0.08    | 1.2 &plusmn;  0.1 |1.57 &plusmn;0.13 |1.99 &plusmn;  0.17 | 2.46 &plusmn;  0.21   |
> > | RIM       | 0.49 &plusmn;  0.02          | 0.7 &plusmn;  0.02    |0.95 &plusmn;  0.03 |1.24 &plusmn;0.04 |1.57&plusmn;  0.06| 1.94 &plusmn;  0.07   |
> > | GPT        |0.58 &plusmn;  0.04          | 0.83 &plusmn;  0.06    |1.13 &plusmn;  0.08 |1.48 &plusmn;0.1 |1.87&plusmn;  0.14| 2.31 &plusmn;  0.17 |
> > | Proposed   | **0.45**&plusmn;  0.01|**0.65**&plusmn;  0.02 |**0.89**&plusmn;  0.03|**1.16**&plusmn;  0.03|**1.47**&plusmn;  0.04|**1.81**&plusmn;  0.05|
> >
> > Consistent with the results on other network topologies in figure 4, our method still outperforms other competing methods on Erdős–Rényi graph.
> >
> > Additionally, we discussed the computational overhead of the proposed method in details, and empirically demonstrated the scalability of our sampling algorithm on large-scale networks (Table 3).
> >
> > We sincerely hope our response adequately addresses your concerns and aids in the evaluation of our work. We look forward to further discussions with you.

---

> ### Comment · Reviewer_emFb · 2024-08-14
> **Thanks for the response and I will maintain my score.**
>
> Thanks for the response and I will maintain my score.

---

> > ### Author Response · Authors · 2024-08-14
> > **Thanks for the comment**
> >
> > We thank the reviewer very much for the feedback. Please do let us know if there is any effect we can futher make to address your concerns. Thank you!

---

### Official Review · Reviewer_4PW9 · 2024-07-13

**Soundness:** 3
**Presentation:** 2
**Contribution:** 3
**Rating:** 6
**Confidence:** 2

**Summary:**

The work proposes an offline/batch active learning method for querying labels for nodes of a graph. The setting assumes access to noisy responses for the subset of nodes queried by the active learner. On the theoretical side, gains are shown over random selection, and bounds are shown on the generalization error of the proposed method. On the empirical side, the proposed algorithms is evaluated against random selection and other relevant baselines, and ablation studies are provided.

**Strengths:**

- Label efficient learning in structured settings is a fundamental learning problem with numerous applications.
- The proposed approach for graph active learning has both theoretical and empirical verified advantages over random selection.
- The meta-approach of balancing informativeness and representativeness can be useful in other contexts.
- Authors provide empirical evaluation which show superiority of proposed approach, and ablation studies which investigate the role of algorithmic components.

**Weaknesses:**

- Missing connections to related theoretical literature on active learning on graphs. E.g. [1, 2]
[1] Dasarathy, G., Nowak, R., & Zhu, X. (2015, June). S2: An efficient graph based active learning algorithm with application to nonparametric classification. In Conference on Learning Theory (pp. 503-522). PMLR.
[2] Zhang, J., Katz-Samuels, J., & Nowak, R. (2022, June). Galaxy: Graph-based active learning at the extreme. In International Conference on Machine Learning (pp. 26223-26238). PMLR.
- Presentation can be made clearer, see suggestions below.

**Questions:**

- What is the significance of assumptions (1) and (2)? (lines 93-94) Specifically, is there a way to evaluate how strong assumption (1) is in practice? Also what is meant by "$\bf f$ is influenced by node covariates $\bf X$"?
- Can you shed more light on the trade-off between informativeness and representativeness? Is there a standard way to quantify these?
- "Robust" in the title is a bit confusing as it may be read as adversarial robustness, while you only seem to consider random noise. Consider using an alternative like "noise-tolerant". Furthermore, it would be good to state the type of noise (independent, bounded variance) considered early on, in Sections 1 and 2 (e.g. lines 58, 85).
- Is there any approximation or other guarantee known for the quality of the proposed greedy approach (line 119) for selecting $\mathcal{S}$ to maximize the threshold frequency?
- Presentation of the greedy algorithm can be improved say using an algobox.
- Line 127: prescence or abscence
- Could you elaborate on the size of $m$ needed in Theorem 2? Can this be empirically estimated?
- What is the running time complexity for the proposed algorithm?
- Is it possible to empirically verify the tightness/looseness of the theoretical bounds?
- Repeated references [2] and [3]

**Limitations:**

Authors should elaborate further on the limitations, e.g. assumptions needed for the theoretical results.

---

> ### Author Rebuttal · Authors · 2024-08-06
>
> **W.1**
> We appreciate the reviewer highlighting relevant literature. Both our method and [1] derive relation between the performance of graph-based active learning and sample complexity. [1] quantifies the complexity on the graph domain of network. In contrast, our method examines the complexity in the spectrum domain. Interestingly, both [1] and our theory suggest query budgets scale approximately linearly with the labelling function complexity given a fixed estimation error. Both [2] and our work provide theoretical comparisons with baseline query methods. [2] utilizes bisection algorithm in [1] to show their method achieves better sample balanceness. Our method offers theoretical comparison with random sampling on information gains.
>
> **Q.1**
> We revise the assumption on the graph signal space for clarity.  Denote $\mathbf{U_d}$ as the first $d$ leading eigenvectors of the normalized graph laplacian, and $X_i, i = 1,\cdots, p$ as the $i$th node-wise feature vector. We assume that there exists $d$ for the target graph signal $\bf{f}$ such that:
> $$  \mathbf{f} \in  \text{Span}( \mathbf{U_d  U_d^T} X_1, \cdots, \mathbf{U_d  U_d^T} X_p ).$$
> Based on this assumption, $\bf{f}$ depends on both network topology and node features. We compare with the function space considered by graph convolutional networks (GCN). It is known [3] that the function space of GCN can be represented as $U\hat{G}U^TX$ with $\hat{G} = \text{diag}(g(\lambda_1),\cdots,g(\lambda_n) )$, where $(\lambda_i)_{i=1}^n$
> are eigenvalues of the normalized graph laplacian, and $g(\cdot)$ denotes polynomial functions. Given the same level of homophily in the GCN space such that $g(\lambda_i) =0, i >d$,  we have
> $$U\hat{G}U^TX \approx \text{Span}( \mathbf{U_d  U_d^T} X_1, \cdots, \mathbf{U_d  U_d^T} X_p )$$
> given $\mathbf{U_dX}$ is full rank. In other words, the proposed function space is almost as large as the GCN space, and can approximate complicated graph signals well.
>
> **Q.2**
> The trade-off can be clearly illustrated by Theorem 3, which can be simplified as:
>
> $$\frac{1}{n}\mathbf{E}_Y\| \hat{\mathbf{f}} - \mathbf{f}\|_2^2 \leq    \mathcal{O}( \frac{r_d}{\mathcal{B}}) +  \mathcal{O}( 1+\frac{r_d}{\mathcal{B}}) \times \text{Bias}, $$
>
> where $\text{Bias} = \big( \frac{1}{n}\sum_{i>d,i\in \text{supp}(\mathbf{f})}  \alpha_i^2 \big)$, $r_d$ is the rank of $\mathbf{U_d X}$ , therefore the increasing function of $d$. A large $d$ lowers representativeness among queried nodes, thereby increaseing
> variance in controlling of condition number ($r_d/\mathcal{B}$), while it reduces $\text{Bias}$ term by including nodes more informative for identifying less smoothed components in $\bf{f}$. We show the trade-off via simulations on **Figure 2** of PDF, which checks MSE under different $d$. Compared to small $d$, both MSE and variance are larger with larger $d$ when query is small, while MSE and variance decrease faster as query increases.
>
> **Q.3**
> We thank reviewer for the thoughtful suggestion! We highlight that robustness is in terms of label noise in both the abstract and introduction, and clearly define label noise in section 2.
>
> **Q.4**
> The approximation guarantee of greedy algorithm can be acheived if the threshold frequency $\omega(S)$ proposed in Theorem 1 satisfies submodularity. A function is submodular if
> $f(S \cup\{v\})-f(S) \geq f(T \cup\{v\})-f(T) $   if  \$S \subseteq T $. Based on [4], if $f$ is submodular, then
> $f(S) \geq(1-1 / e) \cdot f\left(S^*\right)$,  where S is the set obtained via the greedy algorithm and $S^*$ is set as the global maximizer.  We can show that $\omega(S)$ is submodular for star, path, and cycle-shaped networks [5]. One can replace the greedy algorithm with the branch-and-bound algorithm, which has stronger approximation guarantee towards global maximization at the cost of higher computational complexity [6]. In practice, we find the greedy algorithm is good enough to maximize the threshold frequency.
>
> **Q.5**
> We improve presentation by using algobox.
>
> **Q.6**
> We revise "prescence" to "presence".
>
> **Q.7**
> We present the a refined lower bound on $m$. To achieve information gain, $m$ needs to satisfy:
> $$( \frac{n - d_{min} - m}{n - m})^m \times ( \frac{n - d_{min} - m}{n - d_{min}})^{d_{min}} \times \sqrt{d_{min}} < 1,$$
> where $d_{min}$ is the smallest node degree in the network. We elaborate on $m$ derived under different $n$ and $d_{min}$ in **Figure 3** of PDF. The results show that $m$ should be larger when $n$ is larger and $d_{min}$ is smaller. In practice, we can run the biased sampling procedure multiple times with different values of $m$, and set $m$ as the largest one that the condition number of covariance matrix $\tilde{X}^T_{\mathcal{S}}W_S\tilde{X}_{\mathcal{S}}$  is less than a threshold, e.g.,10 based on the rule of thumb for a well-conditioned covariance matrix \cite{Applied_regression}.
>
> **Q.8**
> Please see global response.
>
> **Q.9**
> With $d$ and $m$ fixed, Theorem 3 implies MSE decays at a rate of $\frac{1}{\sqrt{\mathcal{B}}}$.  We demonstrate the relationship between MSE and $\mathcal{B}$ on simulated data in **Figure 1** of PDF. The simulation demonstrates that the order of sample complexity in Theorem 3 matches the empirical results, implying the tightness of the generalization bound in Theorem 3.  Also see global response.
>
> **Q.10**
> We remove repeated references.
>
> >**Ref**
>
> [1]Dasarathy et al. (2015). S2: An efficient graph based active learning algorithm with application to nonparametric classification. COLT.\
> [2] Zhang et al. (2022). Galaxy: Graph-based active learning at the extreme. ICML.\
> [3] Wu et al (2019). Simplifying graph convolutional networks. ICML.\
> [4] Nemhauser et al. (1978). An analysis of the approximations for maximizing submodular set functions. Math. Prog. \
> [5] Chung (1997). Spectral graph theory. Amer. Math. Soc., 1997.\
> [6] Morrison et al (2016). Branch-and-bound algorithms: A survey of recent advances in searching, branching, and pruning. Disc. Opt.

---

> > ### Comment · Reviewer_4PW9 · 2024-08-09
> >
> > Thank you for the detailed response. I do not have further questions, and retain my current score.

---

> > > ### Author Response · Authors · 2024-08-10
> > > **Thank you for your response!**
> > >
> > > We appreciate your valuable feedback, which has greatly improved our paper. If you have any further comments or questions, we would be more than happy to discuss and provide clarification.

---

### Author Rebuttal · Authors · 2024-08-07

We are grateful to all reviewers for their time and insightful feedback. We are encouraged that the reviewers found our work:

1. contributes on significant area with umerous applications (4PW9, hH2h)
2. novel and theoretically solid (4PW9, emFb, hH2h, 7FJA)
3. introduces the trade-off between informativeness and representativeness is useful and generalizable (4PW9)
4. shows superiority on extensive empirical data and synthetic data (4PW9, emFb)
5. easy to implement (hH2h)
6. well present (7FJA)

**Main contributions**:
- propose a new offline graph-based active learning method integrating both network and node features, and robust to random label noise.
- introduce complexity measurement of labeling function and query information gain in the spectral domain. Proposed query strategy aligns with function complexity measurement in the spectral domain, thereby ensuring the learning performance.
- derive the generalization error bound for the proposed method, implying the noval trade-off between informativeness and representativeness in node query.
- conduct extensive empirical and ablation studies to verify the superiority of the proposed method.

We have revised our paper based on reviewers' feedback, which greatly improve our paper. Please find figures and tables in the attached one-page PDF. We summarize major revisions below:

**Empirical studies**
- compare  with multiple SoTA offline and offline active learning methods on five benchmark datasets with varying homophily levels (Table 1)
- compare with offline methods on large scale networks (Table 2)
- compare with offline methods on synthetic data with different network topologies and label noise levels (Figure 4)

**Theory analysis**
- disucss and verify the tightness of convergence rate in Theorem 3 via empirical studies (Figure 1)
- disucss and verify the trade-off between informativeness and representativeness via empirical studies (Figure 2)
- disucss and elaborate the candidate size needed for Theorem 2 (Figure 3)

**Computational cost**
- analyze the time complexity of the proposed method
- investigate running time on benchmark networks with different size and verify the scalability (Table 3)

**General discussion**
- model assumption
- relation with existing methods and theoretical results

**Common questions**

**Experiments**

Based on the reviewers' feedback, we have conducted additional experiments, summarized in the attached PDF as follows:

- **Figure 4**: simulation studies on three synthetic networks with different topologies (Small world property, community structure and scale-free property) under different noise level.
- **Table 1**: experiments on real-world networks with different levels of homophily (Homophily networks: Cora, Citeseer and Pubmed; heterophily networks: Texas and Chameleon).
- **Table 2**: experiments on real-world large-scale networks (Ogbn-Arxiv and Co-Physics).

In these experiments, we compared the performance of the proposed algorithm with SoTA offline methods (RIM [1], GPT [2], SPA [3], FeatProp [4]) and online methods (AGE [5], IGP [6]) for graph active learning. The proposed algorithm achieved the best prediction performance on three synthetic networks with different topologies, the large-scale network Arxiv, the homophily network Cora, and the heterophily network Texas. For the other four datasets, its performance was also competitive. We hope the new experimental results justify the theoretical framework of our algorithm and address the reviewers' concerns.

**Computational cost**

The complexity of our biased sampling method is $\mathcal{O}(n+m+nm+n^3)$ where $n$ is number of nodes, $m\leq n$ is the size of candidate set. When the budget of node label query is $\mathcal{B}$, the total computational cost is then $\mathcal{O}(\mathcal{B}(n+m+nm+n^3))$. The main complexity of the proposed sampling method originates from the SVD operation. When dimension of node feature $p<<n$, we can speed up the informative selection via replacing the SVD by Lanczos algorithm to obtain the $p$th largest or smallest eigenvalues and the corresponding eigenvectors. The time complexity of Lanczos algorithm is $\mathcal{O}(pn^2)$ [7]. Then the complexity of the proposed biased sampling method is $\mathcal{O}(pn^2)$ for single node query. This complexity is comparable to GNN-based network active learning methods since GNN in general has complexity $\mathcal{O}(pn^2)$ in single training update [8].

**Optimality of sample complexity**
When the target graph signal has finite complexity over spectral domain, or fast decay on its heterophily components, then the generalization error in Theorem 3 is dominated by the rarte factor $\mathcal{O}(1/\mathcal{B})$, which matches the optimal linear sample complexity in active learning tasks [9,10]. In addition, the MSE converges at the rate $\frac{1}{\sqrt{\mathcal{B}}}$, which is the optimal nonparametric convergence rate in statistics.

**References**

[1] RIM: Reliable Influence-based Active Learning on Graphs. NeurIPS 2021.

[2] Partition-based active learning for graph neural networks. TMLR 2023.

[3] A Structural-Clustering Based Active Learning for Graph Neural Networks. ISIDA 2024

[4] Active Learning for Graph Neural Networks via Node Feature Propagation. Arxiv 2021

[5] Active learning for graph embedding. Arxiv 2017

[6] Information gain propagation: a new way to graph active learning with soft labels. ICLR 2022

[7] Golub, Gene H., and Charles F. Van Loan. Matrix computations. JHU press, 2013.

[8] Wu, Zonghan, et al. "A comprehensive survey on graph neural networks." IEEE transactions on neural networks and learning systems 32.1 (2020): 4-24.

[9] Chen, Xue, and Eric Price. "Active regression via linear-sample sparsification." COLT. PMLR, 2019.

[10] Dasarathy, et al. "S2: An efficient graph based active learning algorithm with application to nonparametric classification." COLT, 2015.

---

### Comment · Area_Chair_uxNW · 2024-08-12
**technical questions**

Dear Authors and Reviewers,

Based on the current ratings, the paper is at borderline. Since the discussion with authors will end soon, I hope the authors or reviewers could clarify some of my questions so that I can make informed decision. My questions will be mostly theoretical, given the subject of the work.

1. The paper claims using active learning techniques to save labels. What is the obtained label complexity and how does it improve upon passive learning?

2. The paper claims robustness. How is the noise model related to applications and what is the theoretical noise tolerance?

3. In Theorem 2, it is shown that E(Delta) - E(Delta_R) is lower bounded by some quantity. Can you showcase how the lower bound scales under simple assumptions, say the random graph G(n, p) where there are n nodes and each pair is connected with probability p independently?

4. When does the generalization error in Theorem 3 vanish with respect to the sample size, at what rate? Again, can you show concrete results under the G(n, p) model?

Thanks,
Area Chair

---

> ### Author Response · Authors · 2024-08-13
> **Response to AC's comments**
>
> We appreciate AC's insightful theoretical questions! The point-by-point responses are as follows:
>
> **Q1** Denote the eigenvalue distribution $F(\omega)= \frac{1}{n}\sum_{i=1}^n\mathbf{1}(\lambda_i \leq \omega) $ and the eigenvector space $\mathbf{U_d} = \text{Span}(U_1,\cdots,U_d)$, where ${(\lambda_i)}_{i=1}^{n}$, and ${(U_i)}$ are eigenvalues and $d$-leading eigenvectors of the normalized graph laplacian. Assuming the frequency of $\mathbf{f}$ is lower than $\omega_0$,
> we have $\mathbf{f}\in \mathbf{U}_d$ with $d = \lceil nF(\omega_0) \rceil$. Wlog, we assume $r_d = d$ in Theorem 3, i.e, $\mathbf{f}$ has component on each $U_i,i=1,\cdots,d$. With MSE $\frac{1}{n}\mathbf{E}\| \hat{\mathbf{f}} - \mathbf{f}\|_2^2$ fixed, the label complexity of our method is $O(d)$, while that of passive learning is $O(\tilde{d}\log \tilde{d})$ [1] with $\tilde{d} = \lceil nF\big(\frac{n}{\mathcal{B}}F^{-1}(\frac{d}{n})\big)\rceil$ where $\mathcal{B}<n$ is the number of query. Notice that $\tilde{d}>d$ due to the monotonicity of $F(\cdot)$ and $\tilde{d} \rightarrow d$ as $\mathcal{B}$ increases. Our method improves in two aspects: (1) the information selection identifies $\mathbf{f}$ via $O(d)$ queries, and passive learning uses $O(\tilde{d})$ queries, though this improvement decreases as $\mathcal{B}$ increases. (2) Our method improves by a log factor compared to passive learning via actively controlling the condition number of the covariate matrix, therefore the estimation variance.
>
> **Q2** In the real world, data labels aren’t always accurate/reliable. In this paper, we introduce label noise via a probabilistic model $Y_i = \mathbf{f}(i) + \epsilon_i$, where $Y_i$ and $\mathbf{f}(i)$ are the queried and true label on the $i$th node, and $(\epsilon_i)_{i=1}^n$ are independent errors satisfying $\mathbf{E}(\epsilon_i) = 0$ and $\mathbf{Var}(\epsilon_i) = \sigma^2$. Our noise model generalizes to different label types in applications. For continuous labels $Y_i$, one can set $\epsilon_i \sim \text{Normal}(0,\sigma^2)$. For binary label with $\mathbf{f}(i) = -1 \ \text{or} 1$, the typical noise model is $ Y_i = \mathbf{f}(i)$ with prob. $p>0.5$ and $ Y_i = - \mathbf{f}(i)$ with $1-p$. The binary noise model can be reparameterized via our noise model as $Y_i = \text{sign}(Y^*_i)$ where $Y^*_i = \mathbf{f}(i) + \epsilon_i$. The binary $\epsilon_i = 2\mathbf{f}(i)(1-p)$ with prob. $p$ and $\epsilon_i = -2\mathbf{f}(i)p$ with $1- p$, and $\mathbf{E}(\epsilon_i) = 0$ and $\mathbf{Var}(\epsilon_i) = 4p(1-p)$. Similarly, our noise model also accommodates multi-class labels via one-vs-other reparameterization. While most literature consider noise level $\sigma^2$ as constant, our method theoretically guarantees recovery as long as noise level $\sigma^2 = O(\mathcal{B}^\eta)$ with $\eta<1$. This result is useful when human labelling errors increase with workload. Additionally, our node selection strategy is invulnerable to label noise due to the offline nature, and our graph function space alleviates high-frequency noise in node features $\mathbf{X}$ due to network smoothing.
>
> **Q3** For a random graph $G(n,p)$, one has both lower and upper bounds on the node degree with high probability. Specifically, with probability greater than $1 - \exp\big(-np(c-1)^2\big)$, the maximum node degree $d_{max} \leq cnp$ with constant $c>1$. Given $np \rightarrow \infty$, the difference in information gain between the proposed method and random sampling can be simplified as
> $$ \mathbf{E}(\Delta) - \mathbf{E}(\Delta_R) >  O(\frac{1}{ c^3 p^2 n^2}) -\frac{1}{n},$$
> with probability larger than $1 - \exp\big(-np(c-1)^2\big)$. Therefore, the proposed method achieves larger information gain under the network sparsity regime $p = O(n^{-1/2}$), covering a broad range of real networks.
>
> **Q4** Following the same definition as in Q1, and assuming that the frequency of $\mathbf{f}$ is lower than $\omega$, the generalization error in Theorem 3 is
> $$  \frac{1}{n}\mathbf{E}\| \hat{\mathbf{f}} - \mathbf{f}\|_2^2 \leq O\Big(\frac{\text{rank}(\mathbf{U}_d^T\mathbf{X})}{\mathcal{B}}\Big),$$
> where $d = nF(\omega)$, $\mathcal{B}$ is the number of queries, and $\mathbf{X}$ is node covariate matrix. With a fixed complexity of $\mathbf{f}$, the MSE decays at a rate of $\frac{1}{\mathcal{B}}$. The network topology affects the convergence rate through both the size of eigenvector space ($d$) and the alignment between two column spaces ($\mathbf{U}_d^T\mathbf{X}$). For illustration, we can obtain a looser bound as $O\Big(\frac{nF(\omega)}{\mathcal{B}}\Big)$. For random graph, the eigenvalue distribution of graph laplacian $F(\omega;n,p)$ can be analytically represented as the convolution of the standard normal distribution and the Wigner’s semi-circular distribution [2].
>
> [1] Randomized algorithms for matrices and data. Found. and Trend. in Mach. Learn. (2011)\
> [2] Spectral distributions of adjacency and Laplacian matrices of random graphs. Ann. of App. Prob. (2010)

---

### Decision · Program_Chairs · 2024-09-25

**Decision:**

Accept (poster)

**Comment:**

This work studies active querying strategy on graph nodes in order to save label queries. Although reviewers found the empirical study can be expanded, reviewers also agree that the new algorithm and analysis are sound. Authors further clarified in the rebuttal that under certain regimes, the label complexity of the proposed approach is provably lower than that of passive learning (though with only a log factor). Under the classical random graph model, the information gain is provably higher.

Overall, while the saving in labels seems less appealing, the paper studies an important problem and has established a set of guarantees that may inspire a line of follow-up works, such as new algorithms that improve the label complexity and analysis on other graph models.